# Navigating the biopsychosocial landscape: A systematic review on the association between social support and chronic pain

Carlo Matej Rinaudo [1☯*], Maxim Van de Velde[1☯], Arnaud Steyaert[2], André Mouraux[1]

**1** Institute of Neuroscience, Université Catholique de Louvain, Brussels, Belgium, **2** Department of Anesthesiology, Cliniques Universitaires Saint-Luc, Brussels, Belgium

☯ These authors contributed equally to this work.
* carlo.rinaudo@uclouvain.be

## Abstract

The biopsychosocial model is widely used to explain chronic pain conditions. Yet, the role of social aspects including social support is not clear. Literature on social support and chronic pain is still sparse and results inconsistent. The aim of this review is to evaluate the association between social support and different aspects of chronic pain such as pain intensity, pain interference, quality of life, depression and anxiety. We performed a search on Pubmed, Embase, PsycINFO, Cochrane Library, Scopus and CINAHL database entries between January 1, 1980 and May 7, 2024. Two independent investigators selected all longitudinal (prospective or retrospective) and cross-sectional studies in adult populations investigating the effect of perceived social support, social support satisfaction or spousal responses on different aspects of chronic pain (persistent or recurrent pain lasting longer than 3 months). Out of 11,908 queried results, 67 studies met our inclusion criteria. After assessing for risk of bias (adapted version of the "JBI Critical Appraisal Checklist for Analytical Cross-Sectional Studies") and quality of evidence (adapted version of the quality assessment tool of Hawker and al.), 35 studies were included in the final analysis. We found that perceived social support was positively associated with quality of life and negatively associated with depression. Social support satisfaction was negatively associated with depression. Spousal responses were positively associated with pain intensity, pain interference and depression. This review found that, in patients with chronic pain, social support is mainly associated with psychological variables. However, most studies were cross-sectional, and most analyses were correlations. There is a need for higher quality longitudinal studies. The type of social support studied should be clearly defined in every study.

## Introduction

Chronic pain is an extensive and complex medical condition affecting a large amount of people worldwide. In Europe, the overall prevalence of chronic pain is estimated to be around 17–19% [1,2]. In 2006, a survey showed that roughly 60% of individuals with chronic pain reported a reduced capacity or inability to work outside their homes, and approximately 20%

**Data availability statement:** All relevant data are within the manuscript and its Supporting Information files.

**Funding:** This project has received funding from the European Union's Horizon2020 research and innovation programme under grant agreement No 848068.

**Competing interests:** The authors have declared that no competing interests exist.

had experienced job loss due to pain [1]. The survey highlights the urgency of improving recognition, treatment, and management of chronic pain within our health system to enhance the quality of life for the affected individuals and reduce economic burdens on our society.

Chronic pain was formerly defined as pain that endures beyond the expected duration of tissue healing [3]. Therefore, it does not possess the acute alerting mechanism of acute nociceptive pain. Currently, chronic pain is defined as persistent or recurrent pain lasting longer than three months [4]. The chosen period of time has no specific scientific background, but is in alignment with the time periods of other chronic medical conditions and has the advantage of standardizing criteria for research purposes [5].

Over the course of the last century, early definitions of pain that limited the concept as a solely physiological response have been reconsidered. The early work of Harold Merskey underlined the importance of integrating the conflict between physiology and psychology when addressing the definition of pain [6]. In 1978, Merskey chaired the International Association for the Study of Pain (IASP) Subcommittee on Taxonomy, which defined pain as: "An unpleasant sensory and emotional experience associated with actual or potential tissue damage or described in terms of such damage" and emphasized the subjective nature of the phenomenon [7]. In 2020, it was argued that advancements in our comprehension of pain justified a reconsideration of its definition and further expanded the principle as follows: "An unpleasant sensory and emotional experience associated with, or resembling that associated with, actual or potential tissue damage" [8]. These definitions clearly outline the fact that pain is a personal experience and that it must be differentiated from nociception.

A slightly earlier published update of the definition of pain proposed the inclusion of a social component: "Pain is a distressing experience associated with actual or potential tissue damage with sensory, emotional, cognitive and social components" [9]. The taskforce members of Raja et al. considered integrating this concept, but ultimately decided against it. This remains questionable given the extensive data supporting the presence of a social dimension in the integration of pain in humans [10,11] and even non-humans [12,13].

To better incorporate this social dimension, we can look for guidance to the biopsychosocial model, which unifies the biological, psychological, and social dimensions of an individual's life. The basic principle of this model is to consider the experience (of pain) as a cohesive entity and not only as a simple summation of sensory impulses [14]. Part of the reasoning behind suggesting this integration stems from the increasing number of studies exploring the association between social factors and pain, with social support (SS) emerging as a significant influencing factor [15–17].

SS is a multi-faceted and complex concept. It can be defined as a resource encompassing material assistance, such as financial aid, and immaterial help, like emotional support. It is provided to individuals with the expectation of reciprocity, aiming to offer assistance and protection and to mitigate the adverse effects of life stress [18]. Depending on the evaluated construct of SS, it can have beneficial or deleterious effects on health conditions such as chronic pain. For example, it has been shown that chronic pain patients with high levels of SS experience less distress and less severe pain, while receiving support in the form of attention from spouses and solicitousness regarding pain behaviour, is associated with heightened pain severity and increased pain behaviours [19]. It is therefore crucial to clarify which aspects of SS are the most relevant to the aims and context of a specific research question and to define the concept of SS that is used. We decided to focus on three social measures that are often evaluated in social studies on chronic pain: perceived social support (PSS), social support satisfaction (SSS), and spousal responses (SpR). The latter is not a measure of SS per se. Instead, it evaluates responses provided by the spouse to the patient's behaviours. It has been widely studied in the domain of chronic pain and was therefore deemed as important to be included

in our review. It is important to precise that these three measures do not encompass the whole spectrum of SS. Other measures, e.g., network analysis and social connectedness, add a layer of understanding to this complex concept. However, given the scope of synthesis, we could not include every aspect of SS.

PSS must be distinguished from received SS. It is the belief in the availability of this support (PSS), whereas received SS involves its mobilization and expression [20]. A strong psychological sense of support enables individuals to cope, without actively mobilizing network resources. Measures of PSS assess the quality or adequacy of social support from a subjective perspective. PSS can be defined by the type support (emotional, instrumental, informational…) and by whom it is provided (overall, family, spouse…). Higher levels of PSS have been associated in the literature with better mental health [21], physical health, and lower mortality [22].

SSS can be defined as the subjective evaluation of an individual's contentment or fulfilment with the social support they receive from their social network, including family, friends, and community. It is a dimension of SS that reflects the discrepancy among the interactions between real and desired (or necessary) support. Studies have shown an association of SSS with health-related quality of life measures [23], as well as psychological and physical measures of health in patients with fibromyalgia [24].

PSS and SSS are related concepts, but they capture different aspects of an individual's experience with SS. The first one focuses on the individual's beliefs about the availability of support, while latter centres on the individual's assessment of the support they actually receive and their level of contentment with it.

SpR (in the context of chronic pain) refers to the perception (by the patients experiencing chronic pain) of the behaviours and reactions of their spouse or significant other. Partner responses influence pain behaviour and well-being in patients with chronic pain [25]. In chronic pain literature, different types of SpR have been studied. Solicitous responses are characterised by an excessive expression of sympathy and attention to the pain. Punishing responses are defined by the expression of frustration, anger, or criticism towards the individual's pain behaviour. Distracting responses represent an attempt from the partner to divert the individual's attention away from the pain [26]. Each of these responses can yield a different impact on patient's pain behaviour and their well-being. Previous studies have found a relationship between SpR and patients' adjustment to pain, pain intensity, and disability [27,28].

Different models have been developed to explain the effect of SS on health outcomes. Two of them have been widely studied. The 'main' effect model suggests that greater support promotes overall health, reducing the risk of illness. The 'stress-buffering' model suggests that support alleviates stress, lowering the likelihood of illness or expediting recovery after adversity [29]. Despite being older theories, a more recent systematic review [30] presented evidence supporting both the main and the buffering effect. However, the buffering effect was more often able to explain findings in studies that were deemed of higher quality. The evidence suggests that pain reduction, on a social level, is partially mediated by the process of support buffering the adverse influences of stress, through processes such as stress appraisal and active coping. We also have to keep in mind that the two theoretical accounts may not be mutually exclusive.

To our knowledge, very few systematic reviews have explored the association between SS and pain. One study focusing on the impact of informal SS (support provided outside formal/ professional settings) on spinal pain suggested its potential significance as an important factor in the psychological well-being of pain patients, yet findings on the occurrence and prognosis of spinal pain remained inconclusive [31]. While speculative, this study hinted at a potentially greater impact of informal SS in older populations. Another review on the evidence of the effect of SS on pain induced in experimental settings found that explicit expressions of SS (verbally or by handholding) reduced pain, highlighting the significance of intimate relationships

on pain reduction. Although this effect was more prominent in females, no conclusions could be made about a potential sex or gender effect [15]. A more recent systematic review [32] analyzed the relationship between social support and clinical outcomes (pain and disability) in individuals with non-specific chronic low back pain. A small association was found between social support and both pain and disability in people with non-specific chronic low back pain. However, due to missing data, it was not possible to analyze differences according to sex, gender, or type of social support. Common limitations across these reviews included multiple factors of heterogeneity, especially regarding the variety of instruments used to assess SS, and limited sample sizes within the different subgroups of SS.

The present study aims to review the association of PSS, SSS and SpR on the different aspects of chronic pain. More precisely, our objective is to clarify their effect on the development and evolution of chronic pain, on pain intensity, quality of life, and psychological comorbidities of chronic pain. We will also propose suggestions to standardize future studies on the subject in the hope of unifying and facilitating upcoming research in this area.

## Materials and methods

We followed the PRISMA guidelines to perform this systematic review [33]. The study protocol was registered on June 17, 2022, in the international prospective register of systematic reviews (PROSPERO, Ref. CRD42022338899).

### Search strategy

We searched the following databases between January 1, 1980 and May 7, 2024: PubMed, Embase, PsycINFO, Cochrane Library, CINAHL and Scopus. The detailed search strategy can be found in S1 Text. The search strategy yielded 11,908 articles. All of the articles were integrated in Rayyan [34], an online tool to screen articles in systematic reviews.

### Inclusion and exclusion criteria

We used the following inclusion criteria for our study: (1) longitudinal (prospective or retrospective) or cross-sectional studies, (2) English, French or Italian language, (3) peer-reviewed journals, (4) human subjects only, (5) adult population (≥ 18 years), (6) studies including PSS, SSS and SpR and (7) chronic pain (persistent or recurrent pain lasting longer than 3 months) or chronic pain development.

The exclusion criteria were the following: (1) studies not published in full article format or from which data could not be extracted, (2) studies that did not specify their diagnostic criteria for chronic pain, (3) studies that did not include SS, (4) paediatric population (< 18 years).

Since only peer-reviewed articles in full-text format were included, posters, PhD dissertations, or grey literature were excluded. Any study evaluating an association (correlation, regression or path analysis) between SS and one of our outcomes was considered eligible for inclusion in the final analysis. Screening of the identified records was done by two independent reviewers (Dr Rinaudo and Dr Van de Velde) based on the title and abstract (step 1). Following this step, the articles included were screened based on their full text (step 2). Both steps were done in a blind setting. Any disagreement was resolved by discussion or, if needed, by the decision of a third person (Pr Mouraux).

### Data extraction

Data extraction was performed by the two reviewers (Dr Rinaudo and Dr Van de Velde). The following data were extracted: study design, sample size, mean age, type of chronic pain, type of SS, outcomes (Table 1) and direction of the effect (Tables 4–6).

## Risk of bias assessment

Assessing methodological quality (Risk of Bias – RoB) is essential before starting a systematic review [35]. We searched the literature to find the tool best suited for the assessment of the RoB. We opted for an adapted version of the "JBI Critical Appraisal Checklist for Analytical Cross-Sectional Studies" [36]. The tool and its explanation can be found in S2 Text. Briefly, we assessed whether the inclusion criteria were clearly defined, the study subjects and setting were described with sufficient details, explicit criteria for assessing chronic pain were used (through appropriate questionnaires or by a physician), outcomes were measured in a reliable and valid way, and appropriate statistical analysis was used. Each domain was assessed by responding "yes", "no, "unclear", or "not applicable", leading to a final judgement. If the response for each of the five domains was "yes", the study was considered at "low" RoB. If a minimum of three out of the five domains were responded by "yes", the study was considered at "moderate" RoB. A greater importance was given to the item assessing the criteria for chronic pain. If the response for this item was "no", all the other criteria had to be evaluated as "yes" in order for the study to be considered at "moderate" RoB. All other combination of responses were considered at "high" RoB. Only studies at "low" or "moderate" RoB were included in the final analysis, while studies at "high" RoB were excluded. Two independent reviewers (Dr Rinaudo and Dr Van de Velde) were involved in this step. This step was done in a blinded fashion. Any disagreement was resolved through discussion or, if needed, by the decision of a third person (Pr Mouraux).

## Quality of evidence assessment

According to the GRADE system of rating the quality of evidence and grading the strength of recommendation, randomized trials start as high-quality evidence and observational studies as low-quality [37]. The studies included in our review predominantly consist of cross-sectional observational studies and are therefore considered of low-quality evidence according to this system. Nevertheless, assessing the quality of evidence in observational studies remains important. To do so, we opted to use an adapted version of the quality assessment tool of Hawker et al. [38]. The tool and its explanation can be found in S3Text. It is generally used to assess the quality of qualitative studies but was adapted to suit our review. It uses nine items to assess the quality of evidence and provides a score between 9 and 36 points. High-quality studies (grade A) scored 30–36 points, medium-quality (grade B) 24–29 points, and low-quality (grade C) 9–23 points. To ensure we included only the best quality evidence, we opted to only include high-quality studies in the analysis. Two independent reviewers (Dr Rinaudo and Dr Van de Velde) were involved in this step. This step was done in a blinded fashion. Any disagreement was resolved through discussion or, if needed, by the decision of a third person (Pr Mouraux).

Consequently, we included in the final analysis high-quality studies with "low" to "moderate" RoB.

## Data analysis

Due to the heterogeneity in the measurement methods for the different SS constructs, the outcomes, and the lack of data on effect sizes, we could not perform quantitative analyses. The lack of longitudinal studies (eight studies in total, six on PSS and two on SR) and the fact that they evaluated different outcomes, did not allow us to pool these studies together to provide more informative conclusions on causality. Data was synthesised by vote counting, based on the direction of effect whenever there was a statistically significant association. In this review, the term association is used when referring either to a statistical correlation or relationship

**Table 1. Characteristics and results of reviewed studies.**

| Author, Year, Country | Study design | Sample size (f:m), Mean age ± SD | Type of chronic pain | Social support index (scale) | Tested outcomes |
|---|---|---|---|---|---|
| Baker A. et al., 2011, USA | Cross-sectional | N = 247 (180:67) 69.4 ± 9.4 years | Chronic pain | Inventory of Socially Supportive | Depression |
| Bergman S. et al., 2002, Sweden | Longitudinal (3 years) | N = 1852 (?:?) / | Chronic regional and widespread pain | Study specific questionnaire (one item) | Pain intensity (development and persistence of chronic pain) |
| Braunwalder C. et al., 2022, Switzerland | Longitudinal (24 weeks) | N = 343 (89:254) 53.5 ± 0.91 years | Spinal cord injury | Swiss Household Panel [perceived instrumental and emotional social support] | Pain intensity (pain trajectories) |
| Brooks B. et al., 2021, USA | Cross-sectional | N = 419 (401:18) 47.7 ± 13.1 years | Fibromyalgia | Multidimensional Health Profile, Psychosocial Functioning Index | Mental and physical Health-Related Quality of Life |
| Buenaver L. et al., 2006, USA | Cross-sectional | N = 1635 (57% f) 45.8 ± 13.9 years | Chronic pain | Multidimensional Pain Inventory | Pain interference Depressive symptoms |
| Burri A. et al., 2017, Switzerland | Cross-sectional | N = 43 (33:10) 51.8 ± 10.8 years | Chronic pain | German Social Support Questionnaire [perceived emotional and practical social support] | Pain intensity Anxiety |
| Burns J.W. et al., 2020, USA | Longitudinal (3 months) | N = 375 (375:0) / (between 18 and 40 years old) | Persistent pain | Adapted version of the Weiss's Social Provision Scale [perceived social support] | Pain intensity |
| Campos R. P. et al., 2011, Portugal | Cross-sectional | N = 76 (76:0) 49.6 ± 10.1 years | Fibromyalgia | Social Support Satisfaction Scale (ESSS) | QoL (Health-related QoL) Pain Interference |
| Cano A. et al., 2000, USA | Cross-sectional | N = 165 (88:77) 48.6 ± 13.6 years | Chronic pain | Multidimensional Pain Inventory [solicitous – negative – distracting support] Marital Adjustment Test | Pain Intensity Depressive Symptoms |
| Cano A. et al., 2004, USA | Cross-sectional | N = 96 (58:38) 53.3 ± 13.8 years | Musculoskeletal pain | Multidimensional Pain Inventory [solicitous – negative – distracting support] Social Provisions Scales | Pain intensity |
| Chung J. M. et al., 2019, USA | Study 1: Longitudinal (21 days) Study 2: Longitudinal (4 years) | Study 1: N = 220 (195:25) 51.3 ± 11.0 years Study 2: N = 483 (298:185) 55.9 ± 12.6 years | Study 1: Fibromyalgia Study 2: Chronic pain + neurological or neuromuscular disability | Study 1: Satisfaction with Social Support (study specific questionnaire) Study 2: Multidimensional Scale of Perceived Social Support (MSPSS) [perceived social support] | Study 1: Pain intensity (morning) Depressive symptoms Pain interference (afternoon) Study 2: Pain intensity Depressive symptoms |
| Coady A. et al., 2023, Canada | Cross-sectional | N = 305 (226:79) 55.6 ± 13.6 years | Chronic pain | Multidimensional Scale of Perceived Social Support (MSPSS) [perceived social support] | Depression |
| Costello E. et al., 2015, Ireland | Cross-sectional | N = 65 (4:61) 30-49 years: 76.7% | Chronic pain | Multidimensional Scale of Perceived Social Support (MSPSS) [perceived social support] | Pain severity Pain interference Depression Anxiety |
| D'Amico D. et al., 2015, Italy | Cross-sectional | N = 194 (160:34) 43.9 ± 0.9 years | Chronic Migraine | Medical Outcome Study-Social Support Survey (MOS-SSS) [perceived support availability] | Pain interference (pain disability) |
| Dams L. et al., 2022, Belgium | Longitudinal (1 year) | N = 164 (164:0) / | Breast Surgery Pain | McGill QoL support subscale [perceived social support] | Pain intensity |

*(Continued)*

**Table 1.** (Continued)

| Author, Year, Country | Study design | Sample size (f:m), Mean age ± SD | Type of chronic pain | Social support index (scale) | Tested outcomes |
|---|---|---|---|---|---|
| Di Tella M. et al., 2017, Italy | Cross sectional | N = 153 (153:0) 52.4 ± 10.0 years | Fibromyalgia | Multidimensional Scale of Perceived Social Support (MSPSS) [perceived social support] | Pain intensity Depression Anxiety |
| Donaghy B. et al., 2022, United Kingdom | Cross sectional | N = 90 (83:5 + 2 non-binary) 39.1 ± 12.1 years | Complex Regional Pain Syndrome Fibrmyalgia Lower Back Pain Other Chronic Pain | PROMIS Instrumental Support-Calibrated Items v2.0 [instrumental support] | Pain intensity Pain interference |
| Du Plessis M., 2009, South Africa | Cross sectional | N = 31 (31:0) 39.08 ± 12.14 years | Fibromyalgia | Quality of Social Support Scale [perceived quality of social support] | Pain intensity |
| Dybowski C. et al., 2018, Germany | Longitudinal study (12 months) | N = 109 (65:44) 49.3 ± 16.7 years | Chronic Pelvic Pain Syndrome | The 14-item form of the Social Support Questionnaire (F-SozU) [perceived social support] | Pain intensity QoL (CPPS QoL) |
| Dysvik E. et al., 2004, Norway | Cross-sectional | N = 81 (66:15) 47 years | Chronic Pain (musculoskeletal, headaches, abdominal/pelvic, whiplash, neuropathic) | Study specific questionnaire [perceived quantity of social support] | Physical and Mental Health related QoL |
| Edwards R. R. et al., 2022, USA | Longitudinal (6 months) | N = 246 (146:100) 65.1 ± 8.2 years | Total Knee Arthroplasty (TKA) | ENRICH Social Support Instrument (ESSI) [perceived social support] | Pain intensity Pain interference |
| Esteve R. et al., 2021, Spain | Cross-sectional | N = 256 (143:113) 56.5 ± 9.7 years | Chronic Pain | Informal Social Support for Autonomy and Dependence in Pain Inventory (ISSADI) [instrumental – emotional support] | Pain intensity |
| Evers A. et al., 2003, Netherlands | Longitudinal (5 years) | N = 78 (54:24) 57 years | Rheumatoid Arthritis (RA) | IRGL Social Functioning Scales [quantitative and qualitative social support] | Pain intensity |
| Exposito-Vicaino S. et al., 2019, Spain | Cross-sectional | N = 156 (84:72) 61.3 ± 11.7 years | Chronic Cancer Pain | Medical Outcome Study-Social Support Survey (MOS-SSS) [perceived support availability] | Pain intensity Pain interference |
| Faucett J. A. et al., 1991, USA | Cross-sectional | Arthritis N = 84 (69:15) 58.3 ± 13.4 years Myofascial disorders N = 67 (62:5) 47.8 ± 12.0 years | Arthritis Myofascial disorders | Multidimensional Pain Inventory [solicitous – negative – distracting responses] | Pain Intensity Depression |
| Ferreira-Valente M.A. et al., 2014, Portugal | Cross-sectional | N = 324 (214:110) 68.0 ± 15.4 years | Musculoskeletal Pain | Social Support Satisfaction Scale (ESSS) | Pain intensity Pain interference Physical QoL Mental QoL |
| Freitas RPA et al., 2017, Brazil | Cross-sectional | Poor Social Support group N = 17 (17:0) 53.4 ± 7.8 years Normal Social Support group N = 49 (49:0) 52.6 ± 12.5 years | Fibromyalgia | Medical Outcome Study-Social Support Survey (MOS-SSS) [perceived support availability] | Depression |
| Gatien C. et al., 2021, Canada | Cross-sectional | N = 214 (190:24) < 40 years: 36.4% 40-59 years: 52.3% >60 years: 11.2% | Chronic Pain | Dyadic Adjustment Scale [relationship satisfaction] Questionnaire de soutien conjugal [received conjugal support] | Pain intensity Depressive symptoms Anxiety symptoms |

*(Continued)*

**Table 1.** (Continued)

| Author, Year, Country | Study design | Sample size (f:m), Mean age ± SD | Type of chronic pain | Social support index (scale) | Tested outcomes |
|---|---|---|---|---|---|
| Ginting J. V. et al., 2011, Canada | Cross-sectional | N = 180 (0:180) 48.4 ± 10.8 years | Chronic Prostatitis Chronic Pelvic Pain Syndrome | Multidimensional Pain Inventory [solicitous – negative – distracting responses] | Pain interference (pain disability) Physical and Mental QoL Depressive symptoms |
| Glette M. et al., 2018, Norway | Cross-sectional | N = 334 (211:123) 29-44 years: 6.9% 45-64 years: 49.4% > 65 years: 43.7% | Neuropathic Pain Osteoarthritis Spinal Pain | Multidimensional Pain Inventory [solicitous responses] | Pain intensity |
| Gunduz N. et al., 2019, Turkey | Cross-sectional | N = 65 (65:0) 33.5 ± 8.1 years | Fibromyalgia | Multidimensional Scale of Perceived Social Support (MSPSS) [perceived social support] | Pain intensity |
| Goldberg G. M. et al., 1993, USA | Cross-sectional | N = 105 (0:105) / | Chronic Pain | Multidimensional Pain Inventory [solicitous – negative – distracting responses] | Depression |
| Jensen M. P. et al., 2002, USA | Longitudinal (5 months) | N = 61 (19:42) 45.7 ± 13.3 years | Phantom Limb Pain | Multidimensional Pain Inventory [solicitous responses] Multidimensional Scale of Perceived Social Support (MSPSS) [perceived social support] | Pain intensity Pain interference Depression |
| Jeong H. et al., 2020, Korea | Cross-sectional | N = 307 (206:101) / (> 65 years) | Chronic Musculoskeletal Pain | Multidimensional Scale of Perceived Social Support (MSPSS) [perceived social support] | QoL |
| Kerns, R.D. et al., 1990, USA | Cross-sectional | N = 106 (15:91) 51,8 ± 12.8 years | Chronic pain (heterogeneous) | Marital adjustment scale [global marital satisfaction, marital communication] Multidimensional Pain Inventory [support-solicitousness-distracting scales] | Pain severity Depression |
| Kerns R.D. et al., 2002, USA | Cross-sectional | N = 234 (213:21) 50,0 ± 13.8 years | Chronic non-malignant pain | Multidimensional Pain Inventory [support-solicitousness-distracting scales] | Pain intensity Pain disability Depression |
| Kindt S. et al., 2019, Belgium | Longitudinal (14 days) | N = 134 (111:23) 51,7 ± 11.2 years | Chronic pain | Dyadic Coping Inventory [perceived emotional, informational and instrumental social support] | Pain intensity |
| Kovačević I. et al., 2022, Croatia | Cross-sectional | Unsuccessful pain treatment N = 180 (154:26) 62.5 years (range 54.0–71.8 years) Successful pain treatment N = 156 (35:121) 57.0 years (range 46.3–66.0 years) | Chronic non-malignant pain | Self-constructed social support scale (adapted version of the Abbey, Abramis, and Caplan Scale) [perceived social support] | Pain intensity |
| Larbig W. et al., 2019, Germany | Longitudinal (12 months) | N = 52 (11:41) 62.92 ± 2.05 years | Phantom limb pain | Multidimensional Pain Inventory [support-solicitousness-distracting scales] | Pain intensity Depression Anxiety |
| Larice S. et al., 2020, Italy | Cross-sectional | N = 108 (108:0) 53.9 ± 10.3 years | Rheumatoid Arthritis (RA) | Multidimensional Scale of Perceived Social Support (MSPSS) [perceived social support] | Pain intensity Health related QoL |
| Lavin R. et al., 2011, USA | Cross-sectional | N = 163 (51:112) / (> 65 years) | Chronic pain | ENRICH Social support instrument [structural, instrumental and emotional support] | Pain intensity Depressive symptoms |

*(Continued)*

**Table 1.** (Continued)

| Author, Year, Country | Study design | Sample size (f:m), Mean age ± SD | Type of chronic pain | Social support index (scale) | Tested outcomes |
|---|---|---|---|---|---|
| Lee G.K. et al., 2007, Canada | Cross-sectional | N = 171 (84:87) 42.5 ± 9.9 years | Chronic non-malignant pain | The Medical Outcomes Study – Family measure [latent predictor of social and family support] Social Support Index [availability of social support] | Pain intensity Pain interference Depression |
| Lee G.K. et al., 2008, Canada | Cross-sectional | N = 171 (84:87) 42.5 ± 9.9 years | Chronic non-malignant pain | Social Support Index [availability of social support] | Pain intensity Pain interference (impairment) Physical, Psychological and Total QoL Depression |
| Lee S. et al., 2023, South Korea | Cross-sectional | N = 211 (133:78) 72.2 ± 6.0 years | | Social Support Tool [perceived social support] | Health related QoL |
| Leonard M.T. et al., 2018, USA | Cross-sectional | N = 78 (55:23) / | Chronic musculoskeletal pain | Dyadic Adjustment Scale [relationship satisfaction] | Depression |
| Ljungvall H. et al., 2023, Sweden | Cross-sectional | N = 182 (114:67) 51.2 ± 15.8 years | Chronic pain | Multidimensional Scale of Perceived Social Support (MSPSS) [perceived social support] | Pain intensity Pain interference QoL Depression Generalized anxiety |
| López-Martínez A.E. et al., 2008, Spain | Cross-sectional | N = 117 (84:33) 54.0 ± 1.3 years | Chronic pain | Duke-UNC Functional Social Support Questionnaire (Spanish version) [affective support and confidant support] | Pain intensity Depression |
| Matos M. et al., 2017, Portugal | Longitudinal (3 months) | N = 133 (94:39) 78.3 ± 9.1 years | Chronic musculoskeletal pain | Revised Formal Social Support for Autonomy and Dependence in Pain Inventory [perceived promotion of autonomy and dependency] | Pain intensity Pain interference (Pain related disability) |
| Matthias M.S. et al., 2022, USA | Cross-sectional | N = 213 (40:173) 56.8 ± 13.0 years | Chronic musculoskeletal pain | Multidimensional Scale of Perceived Social Support (MSPSS) [perceived social support] | Pain intensity Depression Anxiety |
| Muramatsu N. et al. 1997, Japan | Longitudinal (3 years) | N = 2062 (1115:947) No back pain in 1987 N = 1691 (896:795) 68.9 ± 6.7 years Back pain in 1987 N = 371 (219:152) 69.9 ± 6.8 years | Chronic back pain | Study specific questionnaires on Emotional support (5-point scale) and instrumental support (2-item scale) | Pain intensity (evolution) |
| Nees F. et al., 2022, Germany | Randomized control trial | N = 30 (16:14) With solicitous spouses N = 10 (6:4) 44.4 ± 9.8 years With non-solicitous spouses N = 10 (5:5) 44.4 ± 13.4 years Healthy controls N = 10 (5:5) 46.1 ± 15.2 years | Chronic musculoskeletal pain | Multidimensional Pain Inventory [support-solicitousness-distracting scales] | Pain intensity |
| Nickel J.C. et al., 2008, North America | Cross-sectional | N = 253 (0:253) 45.0 ± 11.3 years | Chronic prostatitis/ Chronic pelvic pain syndrome | Multidimensional Scale of Perceived Social Support (MSPSS) [perceived social support] | QoL |

*(Continued)*

**Table 1.** (Continued)

| Author, Year, Country | Study design | Sample size (f:m), Mean age ± SD | Type of chronic pain | Social support index (scale) | Tested outcomes |
|---|---|---|---|---|---|
| Oraison H.M. et al, 2021, Australia | Cross-sectional | N = 201 (112:89) 47.2 ± 13.4 years | Chronic low back pain | Multidimensional Pain Inventory [support-solicitousness-distracting scales] | Pain intensity Pain interference (Pain disability) |
| Pence L.B. et al., 2008, USA | Cross-sectional | N = 64 (47:17) 42.5 ± 10.2 years | Chronic headache | Spouse Response Inventory [perceived spouse responses to both patients well behaviours and patient pain behaviours] Marital adjustment test [Marital satisfaction] | Pain intensity Pain interference Depressive symptoms |
| Phillips L.J. et al., 2010, USA | Cross-sectional | Multiples sclerosis N = 118 (118:0) 45.5 ± 10.2 years Fibromyalgia N = 197 (197:0) 53.9 ± 9.9 years | Multiples sclerosis and Fibromyalgia | Personnal Resource Questionnaire – Social Support [Intimacy/Assistance, Integration/Affirmation and Reciprocity] | Pain interference (pain disability) Depressive symptoms |
| Piontek K. et al., 2019, Germany | Cross-sectional | N = 234 (131:103) 47.2 ± 17.3 years | Chronic pelvic pain syndrome | The 14-item form of the Social Support Questionnaire (F-SozU) [Self-perceived social support] | Pain intensity QoL |
| Raichle K.A. et al., 2011, USA | Cross-sectional | N = 94 (52:42) 43.2 ± 10.0 years | Chronic pain | Spouse Response Inventory [perceived spouse responses to both patient well behaviours and patient pain behaviours] Multidimensional Pain Inventory [support-solicitousness-distracting scales] | Pain intensity Pain interference (Pain disability) Depression |
| Reich J.W. et al., 2006, USA | Cross-sectional | Fibromyalgia N = 51 (??:??) 51.9 years (range 35–69 years) Osteoarthritis N = 32 (??:??) 58.9 years (range 36–72 years) | Fibromyalgia and osteoarthritis | Modification of the social support items of the scale developed by Manne [partner availability, emotional and instrumental support] | Pain intensity Pain interference (Pain disability) |
| Saarijärvi S. et al., 1990, Finland | Cross-sectional | N = 63 (32:31) 44.0 ± 8.6 years | Chronic low back pain | The marital questionnaire [Marital satisfaction] | Depression Anxiety |
| Smith K. et al., 2015, Australia | Cross-sectional | N = 1418 (794:624) 58 years (IQR = 48-68) No depression N = 519 (265:254) 64 years (IQR = 53–72 years) Pre-pain depression N = 236 (157:79) 55 years (IQR = 44–65 years) Post-pain depression N = 624 (353:272) 55 years (IQR = 47–64 years) | Chronic non-malignant pain | Medical Outcomes Study Social Support Survey (MOS-SSS) [availability of support] | Pain intensity Pain interference Depression Anxiety |
| Solé E. et al., 2020, Spain | Cross-sectional | N = 364 (324:40) 36.3 ± 14.0 years | Chronic pain | Patient-Reported Outcomes Measurement Information System [computed confirmatory factor analysis on instrumental social support, emotional social support, informational social support and companionship] | Pain intensity Pain interference Depressive symptoms |

*(Continued)*

**Table 1.** (Continued)

| Author, Year, Country | Study design | Sample size (f:m), Mean age ± SD | Type of chronic pain | Social support index (scale) | Tested outcomes |
|---|---|---|---|---|---|
| Stroud M.W. et al., 2006, USA | Cross-sectional | N = 70 (25:45) 46.0 ± 11.7 years | Chronic pain in spinal cord injury | Social Support Questionnaire–6 [availability and satisfaction of support] Multidimensional Pain Inventory [support-solicitousness-distracting scales] | Pain intensity Pain interference Depressive symptoms |
| Tripp D.A. et al., 2006, North America | Cross-sectional | N = 253 (0:253) 45.0 ± 11.3 years | Chronic prostatis/ Chronic pelvic pain syndrome | Multidimensional Scale of Perceived Social Support (MSPSS) [perceived social support] Multidimensional Pain Inventory – Solicitous subscale [solicitous responses] | Pain intensity Pain interference (Pain disability) Depressive symptoms |
| Tsai P.-F. et al., 2003, USA | Cross-sectional | N = 71 (54:17) 71.6 ± 7.0 years | Arthritis | Part II of the Personal Resource Questionnaire [perceived social support] | Pain intensity Pain interference (Pain disability) Depression |
| Turk D.C. et al., 1992, USA | Cross-sectional | N = 148 (:) 45.7 ± 13.5 years | Chronic pain | The marital adjustment scale [Marital satisfaction] Multidimensional Pain Inventory [support-solicitousness-distracting scales] | Pain intensity Depressive symptoms |
| Van Alboom M. et al., 2024, Belgium | Cross-sectional | N = 327 (266:61) | Fibromyalgia Secondary chronic pain | Quality of Relationships Inventory [perceived social support] | Pain intensity Pain interference (Pain disability) Depression Anxiety |
| Woods S.B. et al., 2019, USA | Cross-sectional | Baseline acute pain group N = 352 (189:163) 56.1 ± 11.3 years Baseline chronic pain group N = 367 (257:110) 58.8 ± 10.4 years | Chronic non-malignant pain | Relationship Support Family Support Friend Support | Pain persistence |
| Zeng F. et al., 2016, China | Cross-sectional | N = 147 (147:0) 34.9 ± 11.3 years | Chronic pain | Social Support Rating Scale [Subjective Support, Objective Support and Support Availability] | Anxiety |

(regression or path analysis). A distinction was made between the studies showing significant correlations and those showing significant relationships (regressions or path analyses), with the latter providing more evidence for a possible causal link between two variables. Correlations were considered weak when they yielded a Pearson correlation r < 0.4, moderate when 0.4 ≤ r < 0.6 and strong when r ≥ 0.6.

To meaningfully weight results based on the information on article quality, we adapted a levels of evidence tool used in the systematic review of Campbell et al. [31]. The levels of evidence (strong, weak, inconsistent or insufficient) are described in S1 Table and are regrouped based on the review outcomes (see Table 7).

## Results

### Study selection

The search strategy resulted in an initial yield of 11,908 references, of which 5,580 were duplicates. A total of 6,328 were screened (title and abstract), and 199 articles were retained for full-text screening. Out of those, 16 articles could not be retrieved. After full-text screening,

116 articles were excluded ([Fig 1]). Studies were mainly excluded either because they did not evaluate one of the study outcomes (N = 35) or because they did not include chronic pain as their study condition (N = 45). Characteristics of the studies included after the screening process can be found in [Table 1]. One article that was included in the final analysis reported two separate studies [39]. Therefore, 68 studies (published in 67 articles) were assessed for risk

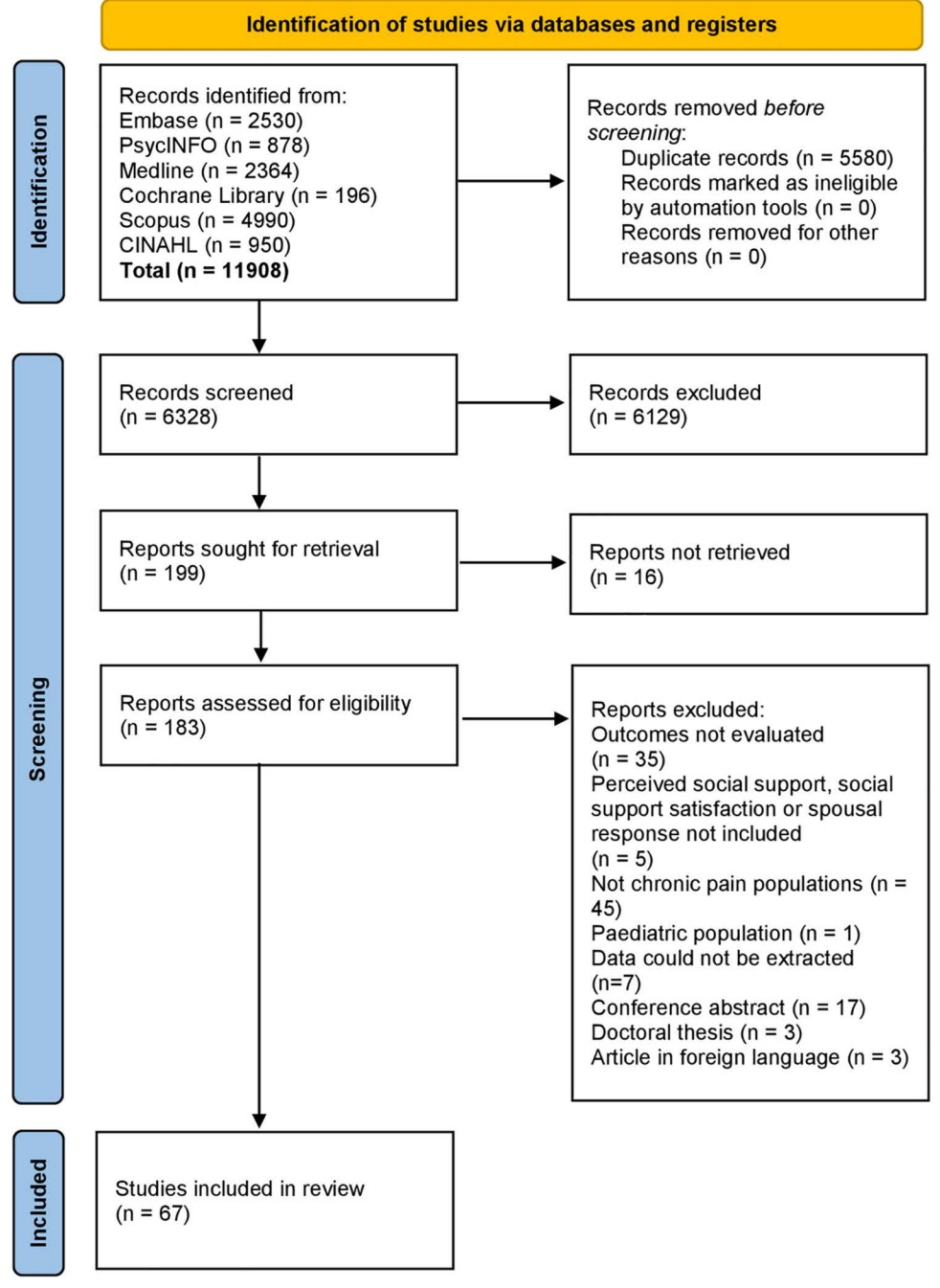

**Fig 1. Prisma flow diagram.**

**Table 2. Risk of bias.**

| STUDY | Inclusion criteria | Study description | Diagnosis criteria | Outcomes measurement | Statistical analysis | RoB |
|---|---|---|---|---|---|---|
| Bergman S. et al., 2001, Sweden | N | N | Y | N | Y | **High** |
| Braunwalder C. et al., 2022, Switzerland | Y | Y | Y | Y | Y | **Low** |
| Brooks B. et al., 2021, USA | N | Y | N | Y | Y | **Moderate** |
| Buenaver L. et al., 2006, USA | N | Y | N | Y | Y | **High** |
| Burri A. et al., 2017, Switzerland | Y | Y | Y | Y | Y | **Low** |
| Burns J.W. et al., 2020, USA | Y | Y | Y | Y | Y | **Low** |
| Campos R. P. et al., 2011, Portugal | Y | Y | Y | Y | Y | **Low** |
| Cano A. et al., 2000, USA | N | Y | U | Y | Y | **Moderate** |
| Cano A. et al., 2004, USA | Y | Y | N | Y | Y | **Moderate** |
| Chung J. M. et al., STUDY 1. 2019, USA | Y | Y | Y | Y | Y | **Low** |
| Chung J. M. et al., STUDY 2. 2019, USA | Y | Y | Y | Y | Y | **Low** |
| Coady A. et al., 2023, Canada | Y | Y | Y | Y | Y | **Low** |
| Costello E. et al., 2015, Ireland | Y | Y | N | Y | Y | **Moderate** |
| D'Amico D. et al., 2015, Italy | Y | Y | Y | Y | Y | **Low** |
| Dams L. et al., 2022, Belgium | Y | Y | Y | Y | Y | **Low** |
| Di Tella M. et al., 2017, Italy | Y | Y | Y | Y | Y | **Low** |
| Donaghy B. et al., 2022, United Kingdom | Y | N | N | Y | Y | **High** |
| Du Plessis M., 2009, South Africa | N | N | Y | Y | N | **High** |
| Dybowski C. et al., 2018, Germany | Y | Y | Y | Y | Y | **Low** |
| Dysvik E. et al., 2004, Norway | Y | Y | Y | Y | Y | **Low** |
| Edwards R. R. et al., 2022, USA | Y | Y | Y | Y | Y | **Low** |
| Esteve R. et al., 2021, Spain | Y | Y | Y | Y | Y | **Low** |
| Evers A. et al., 2003, Netherlands | Y | Y | Y | Y | Y | **Low** |
| Exposito-Vicaino S. et al., 2019, Spain | Y | Y | Y | Y | Y | **Low** |
| Faucett J. A. et al., 1991, USA | Y | Y | Y | Y | Y | **Low** |
| Ferreira-Valente M.A. et al., 2014, Portugal | Y | Y | Y | Y | Y | **Low** |
| Freitas RPA et al., 2017, Brazil | Y | Y | Y | Y | Y | **Low** |
| Gatien C. et al., 2021, Canada | Y | Y | Y | Y | Y | **Low** |
| Ginting J. V. et al., 2011, Canada | N | N | Y | Y | Y | **Moderate** |
| Glette M. et al., 2018, Norway | Y | Y | Y | Y | Y | **Low** |
| Gunduz N. et al., 2019, Turkey | Y | Y | Y | Y | Y | **Low** |
| Goldberg G. M. et al., 1993, USA | Y | N | Y | Y | Y | **Moderate** |
| Jensen M. P. et al., 2002, USA | N | N | Y | Y | Y | **Moderate** |
| Jeong H. et al., 2020, Korea | Y | Y | Y | Y | Y | **Low** |
| Kerns, R.D. et al., 1990, USA | N | N | Y | Y | N | **High** |
| Kerns R.D. et al., 2002, USA | Y | Y | Y | Y | Y | **Low** |
| Kindt S. et al., 2019, Belgium | Y | Y | Y | Y | Y | **Low** |
| Kovačević, I. et al., 2022, Croatia | N | Y | Y | Y | Y | **Moderate** |
| Larbig W. et al., 2019, Germany | N | N | Y | Y | Y | **Moderate** |
| Larice S. et al., 2020, Italy | Y | Y | Y | Y | Y | **Low** |
| Lavin R. et al., 2011, USA | Y | Y | Y | Y | Y | **Low** |
| Lee G.K. et al., 2007, Canada | Y | Y | Y | Y | N | **Moderate** |
| Lee G.K. et al., 2008, Canada | Y | Y | Y | Y | Y | **Low** |
| Lee S. et al., 2023, South Korea | Y | Y | Y | Y | Y | **Low** |
| Leonard M.T. et al., 2018, USA | Y | N | Y | Y | Y | **Moderate** |
| Ljungvall H. et al., 2023, Sweden | Y | Y | Y | Y | Y | **Low** |
| López-Martínez A.E. et al., 2008, Spain | N | Y | Y | Y | Y | **Moderate** |

*(Continued)*

**Table 2.** (Continued)

| STUDY | Inclusion criteria | Study description | Diagnosis criteria | Outcomes measurement | Statistical analysis | RoB |
|---|---|---|---|---|---|---|
| Matos M. et al., 2017, Portugal | Y | Y | Y | Y | Y | **Low** |
| Matthias M.S. et al., 2022, USA | Y | Y | Y | Y | Y | **Low** |
| Muramatsu N. et al., 1997, Japan | N | Y | N | N | Y | **High** |
| Nees F. et al., 2022, Germany | Y | N | Y | Y | Y | **Low** |
| Nickel J.C. et al., 2008, North America | Y | Y | Y | Y | Y | **Low** |
| Oraison H.M. et al, 2021, Australia | Y | N | Y | Y | Y | **Moderate** |
| Pence L.B. et al., 2008, USA | Y | Y | Y | Y | Y | **Low** |
| Phillips L.J. et al., 2010, USA | N | Y | N | Y | Y | **High** |
| Piontek K. et al., 2019, Germany | N | N | Y | Y | Y | **Moderate** |
| Raichle K.A. et al., 2011, USA | Y | Y | Y | Y | Y | **Low** |
| Reich J.W. et al., 2006, USA | N | N | Y | Y | Y | **Moderate** |
| Saarijärvi S. et al., 1990, Finland | Y | Y | Y | Y | Y | **Low** |
| Smith K. et al., 2015, Australia | Y | Y | Y | Y | Y | **Low** |
| Solé E. et al., 2020, Spain | Y | Y | Y | Y | Y | **Low** |
| Stroud M.W. et al., 2006, USA | N | Y | Y | Y | Y | **Moderate** |
| Tripp D.A. et al., 2006, North America | N | Y | Y | Y | Y | **Moderate** |
| Tsai P.-F. et al., 2003, USA | Y | Y | Y | Y | Y | **Low** |
| Turk D.C. et al., 1992, USA | N | N | N | Y | N | **High** |
| Van Alboom M. et al., 2024, Belgium | Y | Y | Y | Y | Y | **Low** |
| Woods S.B. et al., 2019, USA | Y | N | Y | Y | Y | **Moderate** |
| Zeng F. et al., 2016, China | Y | N | Y | Y | Y | **Moderate** |

N = No, Y = Yes, U = Unclear

H = High, M = Moderate, L = Low

of bias (Table 2) and quality of evidence (Table 3). A list of excluded full-text reports can be found in S2 Table.

## Risk of bias and quality of evidence assessment

The summary of the RoB assessment of each study can be found in Table 2. Out of the 68 eligible studies, 8 were deemed at high-RoB (13% of all the included studies), 19 at moderate-RoB (30% of all the included studies) and 41 at low-RoB (57% of all the included studies). Studies at high-RoB most often did not define their inclusion criteria (7 out of 8 studies at high-RoB). Six out of those eight studies did not describe study subjects or settings in detail, and the same number of studies did not use explicit criteria for the assessment of chronic pain.

The summary of the quality of evidence assessment of each study can be found in Table 3. Out of the 68 eligible studies, 6 were of low quality (grade C, 10% of all the included studies), 21 were of medium quality (grade B, 33% of all the included studies) and 41 were of high quality (grade A, 57% of all the included studies). One high-quality study [40] was excluded from the final analysis since it was at high RoB.

The results presented, as well as the discussion, are based on the 40 high-quality with "low" to "moderate" RoB studies included in the final analysis.

## Study characteristics

The total number of participants in the final analysis (40 high-quality studies) of this review is 9481, with the sample sizes per study ranging from 43 to 1418. Female participants

**Table 3. Quality of evidence assessment.**

| Study | Abstract Title | Introduction Aims | Data collection | Sampling | Analysis | Ethics Bias | Results | Transferability | Implications | Total | Grade |
|---|---|---|---|---|---|---|---|---|---|---|---|
| Bergman S. et al., 2001, Sweden | 4 | 4 | 4 | 3 | 3 | 4 | 4 | 3 | 2 | 31 | A |
| Braunwalder C. et al., 2022, Switzerland | 4 | 4 | 4 | 2 | 4 | 4 | 4 | 2 | 4 | 32 | A |
| Brooks B. et al., 2021, USA | 3 | 4 | 4 | 3 | 3 | 3 | 4 | 2 | 3 | 28 | B |
| Buenaver L. et al., 2006, USA | 3 | 4 | 4 | 3 | 4 | 1 | 4 | 3 | 2 | 27 | B |
| Burns J.W. et al., 2020, USA | 4 | 4 | 4 | 3 | 4 | 4 | 4 | 3 | 3 | 33 | A |
| Burri A. et al., 2017, Switzerland | 4 | 4 | 4 | 2 | 4 | 4 | 4 | 2 | 2 | 30 | A |
| Campos R. P. et al., 2011, Portugal | 4 | 4 | 4 | 3 | 3 | 4 | 3 | 3 | 3 | 31 | A |
| Cano A. et al., 2000, USA | 3 | 4 | 4 | 2 | 2 | 1 | 4 | 2 | 4 | 26 | B |
| Cano A. et al., 2004, USA | 4 | 3 | 4 | 4 | 4 | 4 | 2 | 2 | 2 | 29 | B |
| Chung J. M. et al., STUDY 1 2019, USA | 3 | 3 | 4 | 3 | 4 | 4 | 4 | 3 | 3 | 31 | A |
| Chung J. M. et al., STUDY 2 2019, USA | 3 | 3 | 4 | 3 | 4 | 4 | 4 | 3 | 3 | 31 | A |
| Coady A. et al., 2023, Canada | 3 | 4 | 4 | 3 | 4 | 4 | 4 | 3 | 3 | 33 | A |
| Costello E. et al., 2015, Ireland | 3 | 4 | 4 | 3 | 4 | 3 | 3 | 3 | 3 | 30 | A |
| D'Amico D. et al., 2015, Italy | 4 | 4 | 3 | 3 | 3 | 3 | 4 | 3 | 3 | 30 | A |
| Dams L. et al., 2022, Belgium | 4 | 4 | 4 | 2 | 4 | 4 | 4 | 2 | 2 | 30 | A |
| Di Tella M. et al., 2017, Italy | 4 | 4 | 3 | 3 | 4 | 3 | 4 | 3 | 3 | 31 | A |
| Donaghy B. et al., 2022, United Kingdom | 3 | 2 | 3 | 2 | 3 | 4 | 4 | 2 | 3 | 26 | B |

*(Continued)*

**Table 3.** (Continued)

| Study | Abstract Title | Introduction Aims | Data collection | Sampling | Analysis | Ethics Bias | Results | Transferability | Implications | Total | Grade |
|---|---|---|---|---|---|---|---|---|---|---|---|
| Du Plessis M., 2009, South Africa | 3 | 4 | 4 | 2 | 1 | 1 | 3 | 2 | 1 | 20 | C |
| Dybowski C. et al., 2018, Germany | 4 | 4 | 4 | 4 | 4 | 4 | 4 | 4 | 3 | 35 | **A** |
| Dysvik E. et al., 2004, Norway | 4 | 4 | 4 | 3 | 2 | 4 | 4 | 2 | 4 | 31 | **A** |
| Edwards R. R. et al., 2022, USA | 4 | 4 | 4 | 3 | 4 | 4 | 4 | 3 | 3 | 33 | **A** |
| Esteve R. et al., 2021, Spain | 4 | 4 | 4 | 4 | 4 | 4 | 4 | 4 | 3 | 35 | **A** |
| Evers A. et al., 2003, Netherlands | 3 | 4 | 4 | 2 | 3 | 2 | 3 | 2 | 3 | 26 | B |
| Exposito-Vicaino S. et al., 2019, Spain | 4 | 3 | 4 | 2 | 4 | 4 | 3 | 3 | 3 | 30 | **A** |
| Faucett J. A. et al., 1991, USA | 3 | 3 | 4 | 2 | 3 | 1 | 3 | 2 | 3 | 24 | B |
| Ferreira-Valente M.A. et al., 2014, Portugal | 3 | 4 | 4 | 3 | 4 | 3 | 4 | 3 | 3 | 31 | **A** |
| Freitas RPA et al., 2017, Brazil | 3 | 2 | 4 | 2 | 2 | 3 | 2 | 2 | 1 | 21 | C |
| Gatien C. et al., 2021, Canada | 3 | 3 | 4 | 4 | 4 | 4 | 4 | 4 | 4 | 34 | **A** |
| Ginting J. V. et al., 2011, Canada | 4 | 4 | 4 | 2 | 3 | 3 | 3 | 2 | 3 | 28 | B |
| Glette M. et al., 2018, Norway | 3 | 4 | 4 | 3 | 4 | 4 | 4 | 3 | 3 | 32 | **A** |
| Gunduz N. et al., 2019, Turkey | 4 | 2 | 4 | 4 | 3 | 4 | 4 | 2 | 1 | 28 | B |
| Goldberg G. M. et al., 1993, USA | 4 | 3 | 4 | 2 | 1 | 2 | 4 | 2 | 3 | 25 | B |
| Jensen M. P. et al., 2002, USA | 4 | 4 | 3 | 2 | 2 | 2 | 4 | 2 | 3 | 26 | B |
| Jeong H. et al., 2020, Korea | 4 | 4 | 4 | 2 | 4 | 4 | 4 | 2 | 4 | 32 | **A** |
| Kerns, R.D. et al., 1990, USA | 3 | 2 | 3 | 2 | 1 | 2 | 4 | 2 | 2 | 21 | C |

*(Continued)*

**Table 3.** (Continued)

| Study | Abstract Title | Introduction Aims | Data collection | Sampling | Analysis | Ethics Bias | Results | Transferability | Implications | Total | Grade |
|---|---|---|---|---|---|---|---|---|---|---|---|
| Kerns R.D. et al., 2002, USA | 3 | 4 | 4 | 3 | 4 | 3 | 4 | 3 | 4 | 32 | A |
| Kindt S. et al., 2019, Belgium | 3 | 4 | 4 | 3 | 4 | 4 | 4 | 3 | 4 | 33 | A |
| Kovačević, I. et al., 2022, Croatia | 4 | 4 | 4 | 3 | 3 | 4 | 4 | 3 | 2 | 31 | A |
| Larbig W. et al., 2019, Germany | 3 | 3 | 3 | 2 | 4 | 3 | 4 | 3 | 2 | 27 | B |
| Larice S. et al., 2020, Italy | 4 | 3 | 4 | 3 | 4 | 4 | 4 | 3 | 3 | 32 | A |
| Lavin R. et al., 2011, USA | 3 | 4 | 3 | 3 | 3 | 4 | 4 | 3 | 3 | 30 | A |
| Lee G.K. et al., 2007, Canada | 3 | 4 | 4 | 3 | 1 | 3 | 4 | 3 | 3 | 28 | B |
| Lee G.K. et al., 2008, Canada | 3 | 3 | 4 | 4 | 3 | 3 | 4 | 4 | 2 | 30 | A |
| Lee S. et al., 2023, South Korea | 3 | 4 | 4 | 4 | 4 | 4 | 4 | 4 | 3 | 35 | A |
| Leonard M.T. et al., 2018, USA | 4 | 4 | 3 | 2 | 3 | 3 | 3 | 2 | 2 | 26 | B |
| Ljungvall H. et al., 2023, Sweden | 4 | 4 | 4 | 3 | 4 | 4 | 4 | 3 | 3 | 33 | A |
| López-Martínez A.E. et al., 2008, Spain | 3 | 3 | 3 | 3 | 2 | 4 | 4 | 3 | 3 | 28 | B |
| Matos M. et al., 2017, Portugal | 3 | 3 | 4 | 3 | 4 | 3 | 3 | 3 | 4 | 30 | A |
| Matthias M.S. et al., 2022, USA | 4 | 4 | 4 | 3 | 4 | 3 | 3 | 3 | 3 | 31 | A |
| Muramatsu N. et al. 1997, Japan | 3 | 4 | 2 | 3 | 4 | 3 | 4 | 2 | 4 | 29 | B |
| Nees F. et al., 2022, Germany | 3 | 3 | 4 | 2 | 4 | 3 | 4 | 2 | 4 | 29 | B |
| Nickel J.C. et al., 2008, North America | 4 | 2 | 4 | 3 | 3 | 4 | 4 | 3 | 3 | 30 | A |
| Oraison H.M. et al, 2021, Australia | 4 | 4 | 3 | 2 | 3 | 2 | 3 | 2 | 3 | 26 | B |
| Pence L.B. et al., 2008, USA | 3 | 4 | 4 | 3 | 2 | 4 | 3 | 3 | 4 | 30 | A |
| Phillips L.J. et al., 2010, USA | 2 | 3 | 4 | 3 | 4 | 1 | 3 | 3 | 2 | 25 | B |
| Piontek K. et al., 2019, Germany | 4 | 3 | 4 | 3 | 3 | 4 | 4 | 3 | 4 | 32 | A |
| Raichle K.A. et al., 2011, USA | 3 | 3 | 4 | 3 | 4 | 4 | 4 | 3 | 3 | 31 | A |

*(Continued)*

**Table 3.** (Continued)

| Study | Abstract Title | Introduction Aims | Data collection | Sampling | Analysis | Ethics Bias | Results | Transferability | Implications | Total | Grade |
|---|---|---|---|---|---|---|---|---|---|---|---|
| Reich J.W. et al., 2006, USA | 3 | 3 | 3 | 3 | 3 | 4 | 3 | 3 | 4 | 29 | B |
| Saarijärvi S. et al., 1990, Finland | 2 | 3 | 3 | 3 | 2 | 1 | 2 | 3 | 2 | 21 | C |
| Smith K. et al., 2015, Australia | 3 | 3 | 4 | 3 | 4 | 4 | 3 | 3 | 3 | 30 | A |
| Solé E. et al., 2020, Spain | 4 | 4 | 4 | 4 | 4 | 3 | 4 | 4 | 3 | 34 | A |
| Stroud M.W. et al., 2006, USA | 3 | 4 | 4 | 3 | 4 | 4 | 3 | 3 | 3 | 31 | A |
| Tripp D.A. et al., 2006, North America | 3 | 3 | 4 | 3 | 4 | 4 | 4 | 3 | 3 | 31 | A |
| Tsai P.-F. et al., 2003, USA | 3 | 3 | 4 | 4 | 3 | 4 | 3 | 4 | 3 | 31 | A |
| Turk D.C. et al., 1992, USA | 2 | 2 | 3 | 2 | 1 | 1 | 2 | 2 | 2 | 17 | C |
| Van Alboom M. et al., 2024, Belgium | 4 | 4 | 4 | 4 | 4 | 4 | 4 | 4 | 4 | 36 | A |
| Woods S.B. et al., 2019, USA | 4 | 4 | 4 | 2 | 3 | 3 | 3 | 2 | 4 | 29 | B |
| Zeng F. et al., 2016, China | 3 | 2 | 4 | 2 | 3 | 1 | 3 | 2 | 3 | 23 | C |

1 = Very poor 4 = Good

Grade A = high quality study

Grade B = medium quality study

Grade C = low quality study

represented 62.5% (n = 5928) of the total sample size. Studies were mainly conducted in the USA, Canada or Europe. Only three studies [41–43] were conducted outside of these regions (South Korea and Australia). The exact mean age-range could not be extracted, because five studies did not specify the mean age of their population and three studies reported the percentage of participants only by age-range. All studies except two [44,45] had either a majority of participants older than or a mean age above 40 years. Types of chronic pain examined in the studies included unspecified chronic pain conditions, musculoskeletal pain, fibromyalgia, spinal cord injury, neurological/neuromuscular pain or disability, chronic migraine/headache, post-surgical pain (breast surgery), chronic pelvic pain (in both female and male populations) and articular pain.

Out of the 40 high quality studies, 30 of them included a mix of regression analyses or path analyses and correlation analyses. The 10 remaining studies only reported correlation analyses.

Thirty-three studies evaluated PSS through different questionnaires. Two of them evaluated multiple aspects of PSS within the same study [46,47]. Another study evaluated the

impact of PSS in female and male participants separately [41], and one study [48] compared two groups (successful treatment and unsuccessful treatment) to understand which non-medical factors predicted poor outcome of pain treatment in non-malignant chronic pain. One longitudinal study [44] identified psychosocial factors (including PSS) implicated in the transition from acute to persistent pain in women who presented acute pain complaints at the emergency department. Two studies examined the impact of PSS provided by the partner or spouse [49,50]. Overall, the 33 studies yielded 39 analyses on the impact of PSS on the study outcomes (see Table 4). Twenty-three out of the thirty-three studies performed either regression analysis or path analysis on at least one outcome.

Six studies evaluated the association with SSS. Four of them [39,51–53] evaluated SSS while two studies focused on dyadic and marital satisfaction [49,54]. Out of the six studies, four performed either regression analysis or path analysis on at least one outcome (see Table 5).

Five studies evaluated the association with spousal responses. Three of them evaluated the association with three or four different types of responses and the review outcomes within the same study [53–55]. We therefore evaluated a total of 13 different analyses of spousal responses on outcomes (see Table 6). Four out of the five studies performed either regression analysis or path analysis on at least one outcome.

## Study variables scoring

In this review, higher scores of PSS and SSS indicate that the person perceives having a greater amount of support and is more satisfied with the support received, respectively. Higher scores of SpR indicate that the person receives more support or responses of the category evaluated by the questionnaire. For the outcomes, higher scores of pain intensity and interference/disability indicate a greater feeling of having pain and a greater interference/disability generated by pain on daily living/activities, respectively. Higher scores of QoL indicate that the person perceives having a better QoL. Three different aspects of QoL were evaluated in this study: overall (a broad concept incorporating a person's perception of his/her health and other factors), physical (a person's perception of his/her physical health) and mental (a person's perception of his/her mental health). Higher scores for depression or anxiety indicate that the person feels more depressed or anxious.

Positive associations between variables indicate that an increase in score of one variable is associated to an increased score of the other one. Therefore, a positive association with pain intensity or depression implies that patients with stronger SS will experience more pain or depression, while a positive association with mental QoL indicates an improvement in mental well-being of patients. Conversely, negative associations indicate that as one variable's score increases, the other variable's score decreases. In the context of pain intensity or depression, a negative association implies that patients with stronger SS may experience less pain or depression. Similarly, a negative association with mental QoL indicates a decline in the mental well-being of patients.

## Study results

The results of each individual study are presented in Table 4 and Fig 2 for the association of PSS, Table 5 and Fig 3 for the association of SSS, and Table 6 and Fig 4 for the association of SpR. Table 7 summarises the levels of evidence for each combination of type of SS and outcome.

The bar chart on the left represents the direction of the effect of analyses from high quality with low to moderate RoB studies. The bar chart on the right represents the direction of the effect of analyses from all studies.

**Table 4. Impact of PSS on review outcomes.**

| STUDY | Braunwalder C. et al., 2022 | Burri A. et al., 2017 | Burns J.W. et al., 2020 | Chung J.M. et al., 2019 – STUDY 2 | Coady A. et al., 2023 | Costello E. et al., 2015 | D'Amico D. et al., 2015 | Dams L. et al., 2022 | Di Tella M., et al., 2017 | Dybowski C. et al., 2018 | Dysvik E. et al., 2004 | Edwards R.R. et al., 2022 | Esteve R. et al., 2021 | | | | Exposito-Vicaino S.et al., 2019 | Gatien C. et al., 2021 |
|---|---|---|---|---|---|---|---|---|---|---|---|---|---|---|---|---|---|---|
| TYPE OF PERCEIVED SOCIAL SUPPORT | PSS – Instrumental and Emotional | PSS – Emotional and Practical | PSS | PSS (1) | PSS (1) | PSS (1) | PSS – availability (2) | PSS | PSS (1) | PSS – Emotional, Practical and Integration (4) | PSS – Quantity | PSS (3) | PSS – Emotional promotion for autonomy | PSS – Instrumental promotion for autonomy | PSS – Emotional promotion for dependence | PSS – Instrumental promotion for dependence | PSS – availability (2) | Perception of received conjugal support |
| Sample Size Variable | N = 343 | N = 43 | N = 375 | N = 483 | N = 305 | N = 65 | N = 194 | N = 164 | N = 153 | N = 109 | N = 81 | N = 246 | N = 256 | N = 256 | N = 256 | N = 256 | N = 156 | N = 214 |
| Pain Intensity | − | 0 | − | − | 0 | − | | 0 | 0 | 0 | | − | 0 | + | 0 | 0 | 0 | 0 |
| Pain interference/disability | | 0 | | | − | ⊖ | − | | | | | − | 0 | + | + | + | 0 | |
| QoL | | | | | | | | | | 0 | | | | | | | | |
| Physical QoL | | | | | | | | | | | 0 | | | | | | | |
| Mental QoL | | | | | | | | | | | + | | | | | | | |
| Depression | | | | − | − | − | | | − | | | | | | | | | ⊖ |
| Anxiety | | 0 | | | | − | | | − | | | | | | | | | 0 |

PSS: perceived social support

(1), (2), (3), (4): indicates studies using the same questionnaire

0: no correlation. **0**: no relationship.

−: negative correlation. ⊖: negative correlation, but no significantly negative relationship found. ▮: negative relationship.

+: positive correlation. ⊕: positive correlation, but no significantly positive relationship found. ▮: positive relationship.

We report the direction of the effects of statistically significant results. Results are indicated in bold if the effect was found in a regression or path analysis. If not, the direction of effect corresponds to the direction of the correlation between the variables. If a study found significant correlations, but failed to find significant relationships in regressions or path analyses, the result is circled. Colours have been added to improve the readability of the tables (red for negative associations, yellow for no associations, and green for positive associations).

## Pain intensity and perceived social support

We examined 23 studies, yielding a total of 30 analyses between PSS and pain intensity. Four of them [46,50,56] reported weak positive correlations ($0.12 \leq r \leq 0.32$). Conversely, eight analyses found negative associations between PSS and pain intensity. Among them, four used regression models [44,57–59], and two of them were longitudinal studies [44,57]. Braunwalder and her colleagues [57], found that patients with greater support were more likely to be classified in the decreasing pain group rather than in the stable moderate pain group ($\beta = 0.31$, 95% CI [0.02, 0.60]), while Burns and his colleagues [44] found that patients (exclusively women under 40 years of age) benefitting from low social support were more likely to

| Jeong H. et al., 2020 | Kindt S. et al., 2019 | Kovačević I. et al., 2022 | Larice S. et al., 2020 | Lavin R. et al., 2011 | Lee G.K et al., 2008 | Lee S. et al., 2023 | Ljungvall H. et al., 2023 | | Matos M. et al., 2017 | | | Matthias M.S. et al., 2022 | Nickel J.C. et al., 2008 | Piontek K. et al., 2019 | Smith K. et al., 2015 | Solé E. et al., 2020 | Stroud M.W. et al., 2006 | Tripp D.A. et al., 2006 | Tsai P.F. et al., 2003 | Van Alboom M. et al., 2024 | No. of reports on this outcome | Negative association | No association | Positive association |
|---|---|---|---|---|---|---|---|---|---|---|---|---|---|---|---|---|---|---|---|---|---|---|---|---|
| PSS (1) | PSS (1) | Perceived partner support | PSS | PSS | PSS (1) | PSS (3) | PSS - availability | PSS | PSS (1) | PSS – promotion of autonomy | PSS – promotion of dependence | PSS (1) | PSS (1) | PSS – Emotional, Practical and Integration (4) | PSS – availability (2) | PSS | PSS – availability | PSS (1) | PSS | PSS | | | | |
| N = 101 | N = 206 | N = 134 | N = 180 | N = 156 | N = 108 | N = 163 | N = 171 | N = 211 | N = 182 | N = 133 | N = 133 | N = 213 | N = 253 | N = 234 | N = 1418 | N = 364 | N = 70 | N = 253 | N = 71 | N = 327 | | – | 0 | + |
| 0 | 0 | + | 0 | | 0 | 0 | 0 | | 0 | + | + | ⊖ | – | | | | 0 | ⊖ | | | 30 | 8 | 18 | 4 |
| | | | | | 0 | | 0 | | 0 | ⊕ | ⊖ | | | | | 0 | 0 | ⊖ | – | 0 | 20 | 7 | 9 | 4 |
| 0 | + | | | | ⊕ | + | + | | | | | | | | | | | | | | 6 | 0 | 2 | 4 |
| | | | ⊕ | | 0 | | | | | | | | 0 | 0 | | | | | | | 5 | 0 | 4 | 1 |
| | | | ⊕ | | ⊕ | | | | | | | + | 0 | | | | | | | | 5 | 0 | 1 | 4 |
| | | | – | – | – | | – | | | | | – | | | – | – | 0 | – | – | – | 16 | 15 | 1 | 0 |
| | | | 0 | 0 | | | – | | | | | – | | | | | | | | – | 9 | 5 | 4 | 0 |

maintain elevated levels pain after 3 months from the onset ($\eta^2 = 0.42$, $p < 0.01$). The correlation range of the other studies was $0.14 \leq r \leq 0.47$, with only one longitudinal study showing a moderate correlation (39). The 18 remaining analyses did not show any association with any form of PSS.

## Pain intensity and social support satisfaction

Four studies evaluated the association between SSS and pain intensity. Only one longitudinal study [39] found a weak negative correlation (r = -0.28) with pain intensity in patients with fibromyalgia. The three remaining studies did not show any association between SSS and pain intensity.

## Pain intensity and spousal responses

13 analyses from five studies evaluated the association between SpR and pain intensity. Eight analyses show weak positive correlations with pain intensity ($0.12 \leq r \leq 0.38$). Among them, two failed to show a significant relationship in regression models with solicitous responses in chronic headache and chronic prostatitis patients respectively [54,59]. One found a positive relationship between "negative responses to well behaviour" and pain intensity ($\beta = 0.42$,

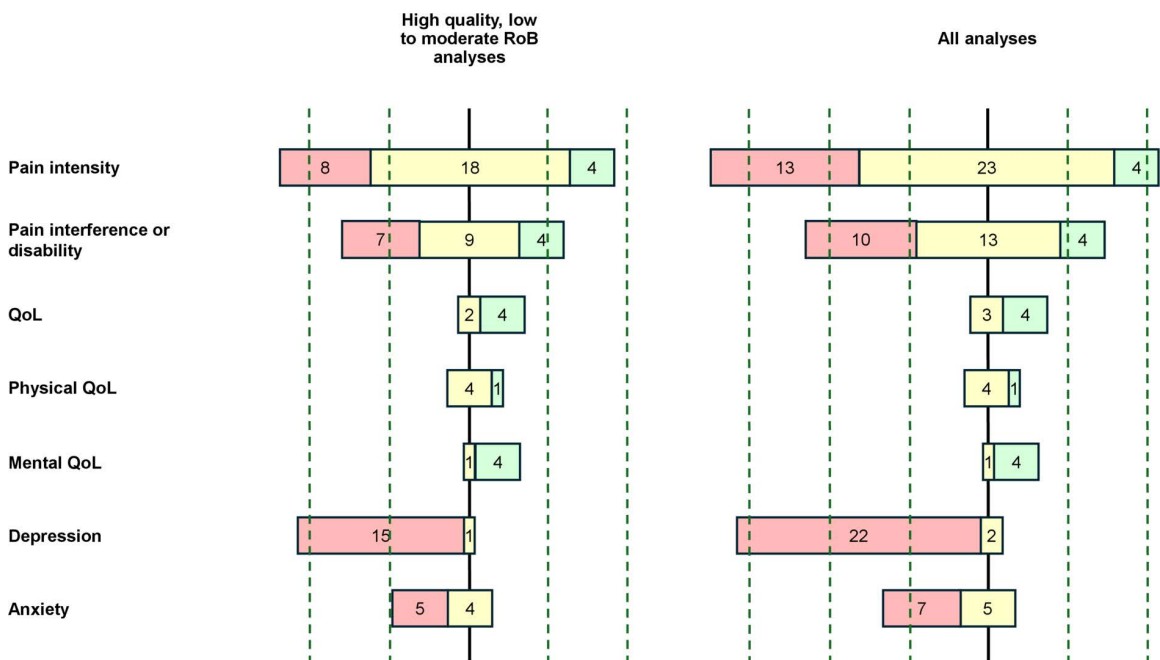

**Fig 2. Bar chart – Association between PSS and review outcomes.** Colour chart: in red, studies that found negative associations, in yellow, studies that found no associations and in green, studies that found positive associations between PSS and the outcome variable. The bar chart on the left represents the direction of the effect of analyses from high quality with low to moderate RoB studies. The bar chart on the right represents the direction of the effect of analyses from all studies.

**Table 5. Impact of SSS on review outcomes.**

| STUDY | Campos R.P. et al., 2011 | Chung J.M. et al., 2019 – STUDY 1 | Ferreira-Valente M.A. et al., 2014 | Gatien C. et al., 2021 | Pence L.B. et al., 2008 | Stroud M.W. et al., 2006 | No. of studies on this outcome | Negative association | No association | Positive association |
|---|---|---|---|---|---|---|---|---|---|---|
| TYPE OF SOCIAL SUPPORT SATISFACTION | SS Satisfaction Scale (1) | **Satisfaction with SS** | SS Satisfaction Scale (1) | Dyadic Satisfaction | Marital Satisfaction | SS Satisfaction | | | | |
| Sample Size Variable | N = 76 | N = 220 | N = 324 | N = 214 | N = 64 | N = 70 | | − | 0 | + |
| Pain Intensity | | − | **0** | | **0** | 0 | 4 | 1 | 3 | 0 |
| Pain interference/ disability | | | **−** | 0 | **0** | **0** | 4 | 1 | 3 | 0 |
| QoL | **0** | | | | | | 1 | 0 | 1 | 0 |
| Physical QoL | **0** | | ⊖ | | | | 2 | 1 | 1 | 0 |
| Mental QoL | **0** | | ⊖ | | | | 2 | 1 | 1 | 0 |
| Depression | | − | | ⊖ | **−** | ⊖ | 4 | 4 | 0 | 0 |
| Anxiety | | | | ⊖ | | | 1 | 1 | 0 | 0 |

SSS: satisfaction with social support

(1): indicates studies using the same questionnaire

0: no correlation. **0**: no relationship.

−: negative correlation. ⊖: negative correlation, but no significantly negative relationship found. ▬: negative relationship.

+: positive correlation. ⊕: positive correlation, but no significantly positive relationship found. ▬: positive relationship.

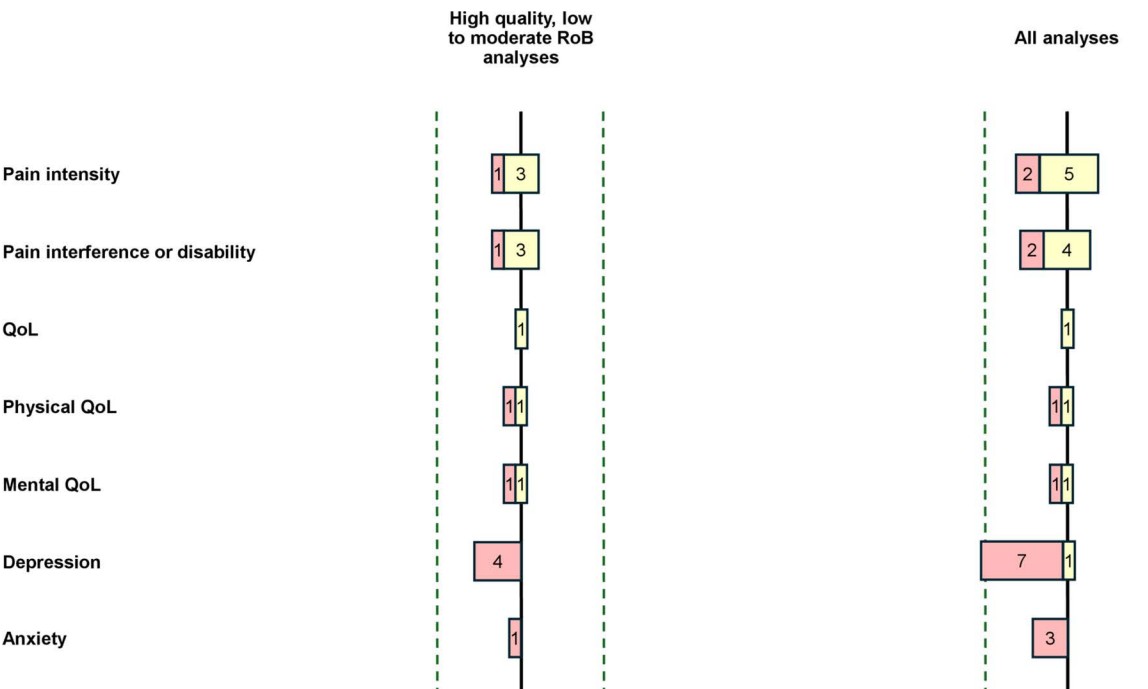

**Fig 3. Bar chart – Association between SSS and review outcomes.** Colour chart: in red, studies that found negative associations, in yellow, studies that found no associations and in green, studies that found positive associations between PSS and the outcome variable. The bar chart on the left represents the direction of the effect of analyses from high quality with low to moderate RoB studies. The bar chart on the right represents the direction of the effect of analyses from all studies.

semi-partial R² = 0.1) in patients with chronic headache [55]. The six remaining studies showed no correlations with pain intensity.

## Pain interference and perceived social support

16 studies yielded a total of 20 analyses between PSS and pain interference. One longitudinal study [47] used a mediational model. No significant results were found, except for a weak positive correlation with perceived promotion of dependence in pain (a subtype of PSS) (r = 0.22). Esteve et al. [46] established correlations between four aspects of PSS: emotional or instrumental promotion for autonomy or dependence and activity impairment. In a previous study, activity impairment was found to be highly associated with pain disability and, therefore, considered in the same pain interference/disability outcome category [60]. All except PSS through emotional promotion for autonomy were positively correlated with activity impairment. Correlations were weak (0.22 ≤ r ≤ 0.36). Seven analyses found negative associations between PSS and pain interference. Four of them used regression models, but only one found a significant relationship between pre-surgery PSS and pain interference (β = -0.02) six months after total knee arthroplasty [61]. Nine studies showed no association between PSS and pain interference.

## Pain interference and social support satisfaction

Four studies evaluated the relationship between SSS and pain interference. One, on chronic musculoskeletal pain, showed a negative relationship (β = -0.23) with SSS explaining 5% of the variance in the outcome [52]. The three remaining studies did not show any association.

**Table 6. Impact of spousal responses on review outcomes.**

| STUDY | Kerns R.D. et al., 2002 | Pence L.B. et al., 2008 | Pence L.B. et al., 2008 | Pence L.B. et al., 2008 | Pence L.B. et al., 2008 | Raichle K.A. et al., 2011 | Raichle K.A. et al., 2011 | Raichle K.A. et al., 2011 | Raichle K.A. et al., 2011 | Stroud M.W. et al., 2006 | Stroud M.W. et al., 2006 | Stroud M.W. et al., 2006 | Tripp D.A. et al., 2006 | No. of studies on this outcome | Negative association | No association | Positive association |
|---|---|---|---|---|---|---|---|---|---|---|---|---|---|---|---|---|---|
| TYPE OF SPOUSAL RESPONSES | PRS – solicitous, distracting, and negative responses | Facilitative responses to well behaviour | Negative responses to well behaviour | Solicitous responses to pain behaviour | Negative responses to pain behaviour | Facilitative responses to well behaviour | Negative responses to well behaviour | Solicitous responses to pain behaviour | Negative responses to pain behaviour | Solicitous responses | Negative responses | Distracting responses | Solicitous Responses | | | | |
| Sample Size Variable | N = 234 | N = 64 | N = 64 | N = 64 | N = 64 | N = 94 | N = 94 | N = 94 | N = 94 | N = 70 | N = 70 | N = 70 | N = 253 | | – | 0 | + |
| Pain Intensity | + | 0 | + | ⊕ | 0 | 0 | + | + | + | 0 | + | 0 | ⊕ | 13 | 0 | 5 | 8 |
| Pain interference/disability | + | 0 | 0 | + | 0 | | | | | ⊕ | + | 0 | ⊕ | 9 | 0 | 4 | 5 |
| QoL | | | | | | | | | | | | | | / | / | / | / |
| Physical QoL | | | | | | | | | | | | | | / | / | / | / |
| Mental QoL | | | | | | | | | | | | | | / | / | / | / |
| Depression | – | 0 | ⊕ | + | ⊕ | ⊖ | 0 | ⊕ | + | 0 | + | + | 0 | 13 | 2 | 4 | 7 |
| Anxiety | | | | | | | | | | | | | | 0 | 0 | 0 | 0 |

0: no correlation. **0**: no relationship.

–: negative correlation. ⊖: negative correlation, but no significantly negative relationship found. ▮ –: negative relationship.

+: positive correlation. ⊕: positive correlation. ⊕: positive correlation, but no significantly positive relationship found. ▮ +: positive relationship.

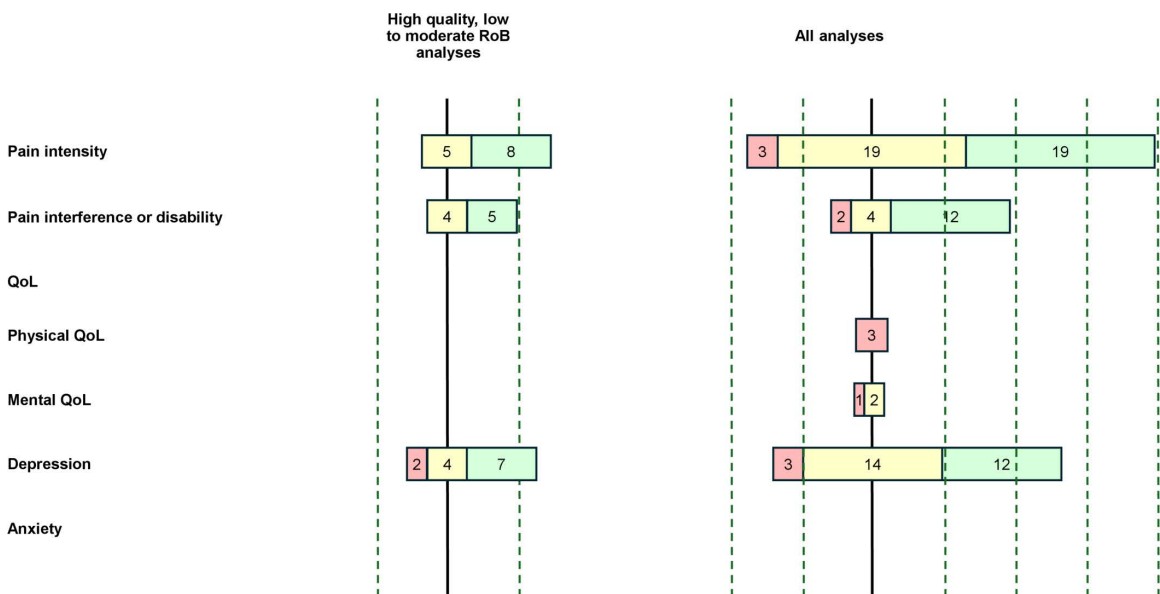

**Fig 4. Bar Chart – Association between SpR and review outcomes.** Colour chart: in red, studies that found negative associations, in yellow, studies that found no associations and in green, studies that found positive associations between PSS and the outcome variable.

Table 7. Level of evidence.

|  | PSS | SSS | Spousal Support/Responses |
|---|---|---|---|
| Pain | Weak for lack of association | Strong for lack of association | Weak for positive association |
| Pain interference | Inconsistent | Weak for lack of association | Inconsistent |
| QoL | Weak for positive association | Insufficient | Insufficient |
| Physical QoL | Strong for lack of association | Insufficient | Insufficient |
| Mental QoL | Strong for positive association | Insufficient | Insufficient |
| Depression | Strong for negative association | Strong for negative association | Weak for positive association |
| Anxiety | Inconsistent | Insufficient | Insufficient |

## Pain interference and spousal responses

9 analyses from four studies evaluated the association between SpR and pain interference. Five yielded positive correlations. Four analyses [53,54,59] attempted to demonstrate a relationship through regression models, with only two obtaining significant results with solicitous responses to pain behaviour (β = -0.36, semi-partial $R^2$ = 0.05) and negative responses (β = 0.33, Total $R^2$ explained by spousal responses = 0.16) [54,59]. The remaining four analyses showed no association between the two variables.

## Quality of life and perceived social support

Six analyses from three studies were reviewed between PSS and overall QoL. In a study on chronic musculoskeletal pain in elderly Koreans, authors found an impact of PSS in women (β = 0.22) but not in men, underlining a possible difference between sexes [41]. In another Korean study with a comparable population [42], the authors developed a structural equation model that supported their hypothesis that PSS had a positive impact on health-related QoL

(direct effect = 0.47, p < 0.05). Another study found a weak positive correlation (r = 0.25) [62]. The remaining study found no association between the PSS and overall QoL of patients suffering from chronic pelvic pain syndrome.

Five analyses from five studies evaluating the links between PSS and physical and mental QoL were reviewed. Physical QoL was weakly and positively correlated to PSS in one study (r = 0.25) [63]. Mental QoL was weakly and positively correlated to PSS in two studies (r = 0.30 and r = 0.37) [62,63]. Two additional studies [64,65] found positive relationships with mental QoL (β = 0.32 and β = 0.24) in a sample of chronic pain patients and a sample of male patients suffering from chronic prostatitis/chronic pelvic pain syndrome. Four studies failed to show any statistically significant relationship with physical QoL; only one failed to do so with mental QoL.

### Quality of life and social support satisfaction

One study on women suffering from fibromyalgia evaluated the link between SSS and QoL (overall, physical and mental) with a linear regression model [51]. SSS was not significant in explaining the variance of all three outcomes. However, the sample size of the study was small (n = 76), and the models may, therefore, have lacked the power to show significant relationships. Another study on chronic musculoskeletal pain patients [52] found weak positive correlations with physical (r = 0.33) and mental (r = 0.39) QoL. However, when included in the regression models, SSS did not explain any of the two outcomes.

### Quality of life and spousal responses

No studies evaluated the link between SpR and overall, physical, or mental QoL.

### Depression and perceived social support

Sixteen analyses on the relationship between PSS and depression were reviewed from sixteen studies. Only one found no association between the two factors [53]. All the other studies found either negative correlations (twelve studies; -0.50 ≤ r ≤ -0.17) or relationships (four studies). These four studies were cross-sectional and focused on chronic pain or chronic musculoskeletal pain. Three of them used regression models and found β values of -0.11 (R² change = 0.073), -0.158 and -0.2. One study conducted a multinomial regression to assess the risk of developing depression in chronic non-cancer pain patients [43]. The results showed an increased risk in developing depression for patients who had lower levels of PSS availability (RRR = 0.84 when comparing patients who had post-pain depression vs those who had not).

### Depression and social support satisfaction

Four analyses from four studies evaluated the association between SSS and depression. SSS was exclusively negatively associated with depression. One study [39] performed only a correlation analysis between the two factors and found a weak negative correlation (r = -0.30). Two other studies found weak negative correlations (r = -0.22 and r = -0.30) but failed to find any relationship in regression models [49,53]. Finally, one study found a negative relationship (β = -0.47, semi-partial R² = 0.13) with marital satisfaction in chronic headache patients [54].

### Depression and spousal responses

13 analyses from five studies between different SpR and depression were analysed in this review. Seven analyses found positive associations with solicitous, negative, and distracting responses (either in general or to both well and pain behaviour). Three of these analyses failed

to find any statistically significant relationship and only found weak and positive correlations ($0.28 \leq r \leq 0.32$). The four other analyses found significant relationships with β values ranging from 0.23 to 0.39. Two analyses found negative correlations ($-0.29 \leq r \leq -0.13$) with pain-relevant support and facilitative responses to well behaviour. One study included SpR in regression models but failed to find any significant relationship [55].

### Anxiety and perceived social support

Nine analyses from seven nine conducted analyses between PSS and anxiety. Four of them found no correlation or relationship, four found weak to moderate negative correlations ($-0.40 \leq r \leq -0.15$) and one [58] found a negative relationship (β = -0.07, $R^2$ change = 0.042).

### Anxiety and social support satisfaction

Only one study evaluated the association between SSS (dyadic satisfaction) and anxiety [49]. A weak and negative correlation was found (r = -0.19) but SSS was not able to significantly explain anxiety in the regression model.

### Anxiety and spousal responses

No studies evaluated the link between SpR and anxiety

## Discussion

Our systematic review identified 40 high-quality studies with "low" to "moderate" RoB, 8 of which were longitudinal. None of the studies were randomized. Overall, the results suggest that SS has a beneficial effect on patients with chronic pain, especially on mental health. Patients reporting greater PSS and SSS tend to score higher on mental QoL and lower on depression scores. Conversely, SpR are associated with increased pain intensity, pain impairment and depressive symptoms.

It is important to mention that all the measures of SS in this review are subjective measures from the patient's point of view. Consequently, such reports are influenced by the way the respondent perceives and processes the information.

### Perceived social support

This review was confronted with many different aspects of SS. Moreover, each were assessed with different questionnaires. In the 33 studies evaluating PSS, 18 different questionnaires were used, and a lot of different aspects of PSS were evaluated (instrumental, emotional or practical support, integration, availability or quantity of support, promotion for autonomy or dependence, spousal/partner or overall PSS). This heterogeneity, both in the studied SS constructs and in the way each was evaluated, could explain some of the inconsistencies found in our review.

Results regarding the association of PSS with pain intensity and interference were inconsistent. One interesting study evaluated pain trajectories over a span of 24 weeks in patients that suffered from spinal cord injury (traumatic or non-traumatic) [57]. Patients were classified in pain trajectory groups. The authors found that patients with better PSS were more likely, at 1 month after injury, to be in the decreasing pain group than the stable moderate pain group. Evolution of pain in the period after its onset is a predictor of chronification over time, especially in traumatic or surgical situations. Identifying predictors of positive evolution is therefore essential to prevent chronic pain. Unfortunately, no similar studies were found in this review. Replication of such studies are essential to help clinicians recognize which patients

are at risk of developing persistent post-traumatic or post-surgical pain. Another study also longitudinal in nature, investigated factors contributing to chronification of chronic pelvic pain [66]. The study showed a marginally significant relationship between baseline PSS and pain intensity at one year. Patients perceiving higher levels of support at baseline tended to report having more pain at 12 months (p = 0.05). As shown by these two studies, results can be contradictory. Based on the results of this review on pain intensity and pain interference, no conclusions can thus be drawn.

However, grouping studies based on the questionnaires that were used to assess PSS yielded interesting trends. Two studies [46,47] used the "Informal Social Support for Autonomy and Dependence in Pain Inventory" questionnaire [67], which is specific to chronic pain conditions. Their results show that the constructs of promotion of autonomy and dependence are either positively correlated to pain intensity (3/6 analyses) and pain interference (4/6 analyses) or show no association. In contrast, all other studies used questionnaires that were not specifically developed for chronic pain patients. For example, the ten studies (eleven analyses) using the "Multidimensional Scale of Perceived Social Support" [68] found either negative associations or no associations with outcomes such as pain intensity (5 analyses with negative associations and 6 with no associations) and pain interference (3 analyses with negative correlations and one with no association).

Such discrepancies suggest that when studying a particular phenomenon, researchers should be mindful of the tools they use and whether they are adapted to the research question. Using unspecific PSS questionnaires might explain the (lack of) results in our review. Alternatively, we could hypothesise that PSS does not have a direct association with "somatic outcomes", but does have an indirect one through its association with other factors, such as psychological ones. It is also possible that the need for SS may vary depending on the chronic pain condition and other factors. To our knowledge, no study evaluating this has been conducted to date.

Regarding overall QoL, the six analyses reported weak evidence for positive association. One study [41] found an improvement of QoL in elderly Korean women who had higher PSS but not in men, suggesting that there might be sex or gender differences in the effect of SS on pain-related outcomes. Future studies should, when possible, attempt to analyse sex or gender dependency of the impact of SS on QoL, as such differences could help tailor health-related strategies for patients.

Regarding physical QoL, none of the studies found a significant relationship with PSS (one study reported a positive correlation, but when they included PSS in a regression model, no relationship was found). The results suggest that these two variables are not associated.

Conversely, we found that PSS is mainly positively associated with mental QoL (4/5), suggesting that patients with higher PSS have an increased mental QoL.

Despite the results, we need keep in mind that the number of studies evaluating the association of PSS and any form of QoL is small (four studies for overall QoL and five studies for each physical and psychological QoL). More studies are required to confirm these findings.

The most probing association of PSS found in this review was with depression. All but one of the 15 studies found negative associations. Four found negative relationships. One was an exposure study evaluating the factors that are associated with the development of depression in chronic non-cancer pain following the onset of opioid treatment [43]. The study found that lower levels of perceived availability of SS was a risk factor for developing depression after the onset of pain, regardless of the onset of opioid treatment. Past studies have stated the importance of treating depression in chronic pain management [69,70]. The results from this review suggest that clinicians should consider the social entourage of the chronic pain patient when tackling depressive symptoms. Working on improving their perception of SS, e.g. through

cognitive biases, or working on the relationship with the patient's entourage, might be an innovative approach to improve depressive symptoms in this population.

Regarding anxiety, results are inconsistent with five out of nine analyses showing negative correlations and one study [58] showing a negative relationship in patients suffering from chronic musculoskeletal pain.

Overall, our results show that PSS is positively associated with chronic pain patient's mental health. From a holistic perspective, it suggests that evaluating the effect of PSS only on somatic outcomes, such as pain intensity, does not capture the full picture of the patient's experience. PSS can have an effect on mental health, which is beneficial regardless of its effect (or lack of effect) on somatic outcomes. Future studies should focus on the influence of PSS on mental health and how it affects the pain experience.

## Social support satisfaction

Regarding SSS, this review analysed six studies. Only two studies used the same questionnaires, and two other studies evaluated either dyadic support satisfaction or marital support satisfaction. Most studies showed an absence of association with pain intensity (3/4) and pain interference (3/4). Regarding QoL (overall, physical and psychological) and anxiety, there were not enough studies to draw any conclusion ($\leq$ 2).

Four studies evaluated how SSS influenced the level of depression. Three of them found negative correlations, and one cross-sectional study on chronic headache [54] found a significantly negative relationship. These results suggest that there might be an association between SSS and depression in patients with chronic pain, although more studies are needed to strengthen this conclusion. Due to the absence of longitudinal studies, we do not have information on causality links between the two variables.

## Spousal responses

SpR yielded opposite results compared with PSS and SSS. Patients reporting greater SpR tended to also report greater pain intensity, pain interference/disability, and greater depression. This is consistent with previous reviews [71,72]. There were no studies evaluating the association between SpR and QoL or anxiety.

These results should not lead to the conclusion that SpR are necessarily detrimental to the health of patients with chronic pain. The type of SpR evaluated by the study will influence the outcome. Some SpR, such as solicitous responses, have been widely studied in the chronic pain literature and tend to reinforce pain behaviours in patients. Researchers have explained this phenomenon through the operant conditioning theory of chronic pain. This theory posits that pain behaviours, while they may initially relate to the actual pain felt, are maintained by the environment (notably by spouse responses) via a process of operant learning after the termination of nociceptive stimulation [73]. For example, the benefit of receiving more attention, sympathy or assistance in response to pain behaviours (solicitous responses), may inadvertently reinforce the expression of such behaviours. Interestingly, Leonard et al. (2006) stated that marital satisfaction might in fact moderate the relationship between spouse solicitousness and the pain experience [72]. According to the author, it would be possible, in the context of a poor relationship, that patients with chronic pain may interpret solicitous responses from spouses in a negative manner or as something spouses feel obligated to do. On the other hand, one could suppose that patients experiencing and expressing greater pain might prompt increased spousal support. Due to the cross-sectional design of the available studies, we were not able to determine the directionality of the association between SpR and pain-related outcomes. As for the other SS constructs, longitudinal studies are required to better understand

the causality link between spousal responses and pain experience. Gaining a deeper knowledge of spousal responses is crucial. It could help develop new chronic pain management trajectories that include the spouse.

In addition, future research needs to focus on creating psychometric tools to assess SS in the domain of chronic pain. Almost none of the questionnaires used to assess PSS and SSS were specific to chronic pain conditions. The questionnaire that was used the most to evaluate PSS was the "Multidimensional Scale of Perceived Social Support" [68] (10 studies). It evaluates, through 12 items, the perceived adequacy of SS through three sources: family, friends and significant other; using a 5-point Likert scale. Regarding SR, only two questionnaires specific to pain conditions were used in studies this review: the "West Haven-Yale Multidimensional Pain Inventory" [27] (WHIMPI) and the "Spouse Response Inventory" [74] (SRI). Both tools have been validated and evaluate different spousal responses to patients suffering from chronic pain. The WHIMPI focuses on solicitous, negative and distracting responses), but does not differentiate between spouse responses that are potentially positively or negatively reinforcing. The SRI (39-item inventory), on the other hand, provides a better understanding of chronic pain patient-spouse interaction by measuring spouse responses not only to pain behaviours but also well behaviours [74]. For these reasons, we recommend using the SRI in future studies evaluating the chronic pain patient-spouse interaction.

## Studies not included in the final analysis

A systematic review aims to provide a thorough and unbiased synthesis of the existing evidence. Given the qualitative nature of our subject, we aimed to preserve credibility and undermine scepticism by excluding studies with a high risk of bias or a low to medium quality of evidence from our final results. Low-quality studies are more likely to produce unreliable results and may not be representative of the general population. The inclusion of such studies could compromise the overall validity and generalizability of our findings. Studies with a high risk of bias often have methodological weaknesses that can introduce confounding variables and may produce misleading or exaggerated results. The inclusion of these biased studies could lead to inaccurate conclusions and distort the overall understanding of the topic. Studies excluded for the above-mentioned reasons can be found in S3-5 Tables.

Of note, four of the included studies had chronic pain development as an outcome. This was an outcome that we sought to analyse originally, but, as these four studies did not meet our quality and risk of bias criteria, they are only reported in S3-5 Tables.

## Study limitations

This systematic review has limitations related to the studies included as well as the review process.

First, the heterogeneity of the reviewed data does not allow to draw definite conclusions. SS and its different constructs were evaluated through many different questionnaires. This may have had an impact on the results that were analysed. In addition, the lack of reported effect sizes in most of the studies only allows us to conclude that there is a tendency in the effect of the predictor on the outcome. The design of the studies was a limiting factor as well. Most studies were observational and cross-sectional. According to the GRADE system to evaluate the quality of evidence in systematic reviews, the quality of non-randomized studies is, by definition, low. Because there were few longitudinal studies, no causal link between variables could be established. As mentioned before, the quality of evidence assessment tool that we

used was originally designed for qualitative studies. After evaluation of the tool, we deemed that it was appropriate to slightly adapt it for the studies included in our review. We adapted the data analysis question, to make it applicable for all type of studies and not qualitative ones exclusively. The RoB tool used in this study also had some limitations since it was constructed for cross-sectional studies. Once again, we assessed and adapted it beforehand and deemed the tool appropriate. The level of evidence summary (Table 7) used a home-tailored method, and serves mainly for the reader to quickly understand whether an association with the outcomes was found and whether the association was deemed "weak" or "strong". It is not an evidence-based medicine tool, and one should not deduce any sort of "grade of recommendation" from it.

Although not a limitation, it is important to state that despite including studies on adults aged of 18 or older, most studies had a mean age > 40 years. One study [57] included patients aged above 16 years. After contacting the author, she confirmed that very few patients included in the study were younger than 18. Since the study fulfilled the other criteria and the vast majority of the patients were adults, this study was still retained. Studies were mainly conducted in Western countries, with only two studies conducted outside of Europe or North America. The results of this study may, therefore, not be generalised to an adult population under 40 years of age and may be representative only of patients from Western countries.

In the study protocol we stated that we would assess the link between SS with chronic pain outcomes. After having included the studies in the final analysis, we made a slight deviation by dividing SS into PSS, SSS, and SpR. Concerning the outcomes, we decided to include pain disability to the outcomes and pool it with pain interference. The decision was taken after analysing the concept of these two outcomes in the included studies. Since they were very similar and, sometimes, even overlapping, we pooled them together.

## Conclusions

We found 40 high-quality studies at "low" or "moderate" RoB evaluating three aspects of social support: perceived social support, social support satisfaction and spousal responses. Studies evaluating perceived social support and social support satisfaction showed a lack of association with pain intensity and inconsistent associations with pain interference/disability. Conversely, patients with higher perceived social support and social support satisfaction reported better scores on psychological outcomes, such as lower reported depressive symptoms and higher mental health quality of life.

Patients reporting greater spousal responses (solicitous, distracting and negative) also reported having more severe symptoms of pain intensity, interference/disability and depression. These results underline the importance of addressing the spouse when considering chronic pain management. The interactions between the patient and the spouse should be investigated by health practitioners. Educating both the patient and the spouse on pain behaviours and spouse responses could help the couple to adapt their interactions and responses.

Given the association between perceived social support and better psychological health outcomes, future interventions could target enhancing the psychological aspects of social interactions. Cognitive-behavioural therapy could be modified to include components that focus on improving perceived social support and satisfaction within relationships.

Developing educational programs for both patients and their spouses could help them understand the impact of different support behaviours on pain perception and psychological well-being. Including spouses in treatment sessions may foster better understanding and

adaptation, helping them to learn effective ways to support their partner and recognize the boundaries of their involvement.

We also suggest routine screening for psychological distress in patients with chronic pain and evaluation of the quality and type of social support these patients experience. This could lead to a more personalized care approach in future pain counselling.

Future studies should focus on implementing longitudinal designs to better understand the nature of the link between these variables. For example, it would be interesting to elucidate whether patients expressing greater pain prompt an increase in social support or vice-versa. Studies should implement questionnaires on social support that have been validated or at least already used in previous studies on chronic pain. To improve conceptual and methodological consistency, future studies should clearly define the type of social support evaluated. This is crucial, since different dimensions of social support may influence health outcomes through different pathways [75].

## Supporting information

**S1 Text. Research Equations.** The research equation was carried out in six different databases (PubMed, Embase, PsycINFO, Cochrane Library, CINAHL and Scopus). The equations were constructed around three main concepts:" adult", "chronic pain" and "social support". Variations between research strings from one database to another are linked to the specificity of the thesaurus of each database.
(DOCX)

**S2 Text. Adapted version of the "JBI Critical Appraisal Checklist for Analytical Cross-Sectional Studies.**
(DOCX)

**S3 Text. Adapted version of the "Quality Assessment for the Systematic Review of Qualitative Evidence.**
(DOCX)

**S1 Table. Adapted level of evidence from Cambell et al. EJP. 2011.** In the vote count of statistically significant associations, we decided to give more weight to significant relationships. Therefore correlations count as one, while relationships count as two.
(DOCX)

**S2 Table. Excluded full-text articles.**
(DOCX)

**S3 Table. Impact of PSS in screened studies that were excluded from the final analysis.** PSS: perceived social support, 0: no correlation. **0**: no relationship., ▬: negative correlation. ⊖: negative correlation, but no significantly negative relationship found. ▪: negative relationship., +: positive correlation. ⊕: positive correlation, but no significantly positive relationship found. ▪: positive relationship.
(DOCX)

**S4 Table. Impact of SSS in screened studies that were excluded from the final analysis.** SSS: social support satisfaction, 0: no correlation. **0**: no relationship., ▬: negative correlation. ⊖: negative correlation, but no significantly negative relationship found. ▪: negative relationship., +: positive correlation. ⊕: positive correlation, but no significantly positive relationship found. ▪: positive relationship.
(DOCX)

**S5 Table. Impact of spousal responses in screened studies that were excluded from final analysis.** 0: no correlation. **0**: no relationship., ⚊: negative correlation. ⊖: negative correlation, but no significantly negative relationship found. ▬: negative relationship., ✚: positive correlation. ⊕: positive correlation, but no significantly positive relationship found. ⊞: positive relationship. (DOCX)

## Acknowledgments

We thank Marie Longton (UCLouvain) for providing help in assembling the research equation.

C.M. Rinaudo and M. Van de Velde are joint first authors.

## Author contributions

**Conceptualization:** Carlo Matej Rinaudo, Maxim Van de Velde.

**Data curation:** Carlo Matej Rinaudo.

**Formal analysis:** Carlo Matej Rinaudo, Maxim Van de Velde.

**Funding acquisition:** André Mouraux.

**Investigation:** Carlo Matej Rinaudo, Maxim Van de Velde.

**Methodology:** Carlo Matej Rinaudo, Maxim Van de Velde.

**Project administration:** Carlo Matej Rinaudo, Maxim Van de Velde.

**Supervision:** Arnaud Steyaert, André Mouraux.

**Writing – original draft:** Carlo Matej Rinaudo, Maxim Van de Velde.

**Writing – review & editing:** Carlo Matej Rinaudo, Maxim Van de Velde, Arnaud Steyaert, André Mouraux.

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
