## [Decision Letter · Decision Letter 0]

17 Sep 2024

PONE-D-24-21939

Navigating the Biopsychosocial Landscape: A Systematic Review on the Association between Social Support and Chronic Pain

PLOS ONE

Dear Dr. Rinaudo,

Thank you for submitting your manuscript to PLOS ONE. After careful consideration, we have decided that your manuscript does not meet our criteria for publication and must therefore be rejected.

I am sorry that we cannot be more positive on this occasion, but hope that you appreciate the reasons for this decision.

Kind regards,

Ashish Wasudeo Khobragade, MD

Academic Editor

PLOS ONE

Additional Editor Comments:

After thoroughly evaluating the manuscript titled “Navigating the Biopsychosocial Landscape: A Systematic Review on the Association between Social Support and Chronic Pain” and considering several factors like novelty, methodological robustness, the potential for impact on policy or practice, reviewers’ comments and publication criteria of the journal, unfortunately, on this occasion, I found it not suitable for publication.

Reviewers' comments:

Reviewer's Responses to Questions

**Comments to the Author**

1. Is the manuscript technically sound, and do the data support the conclusions?

Reviewer #1: Yes

Reviewer #2: Partly

2. Has the statistical analysis been performed appropriately and rigorously? 

Reviewer #1: Yes

Reviewer #2: Yes

3. Have the authors made all data underlying the findings in their manuscript fully available?

Reviewer #1: Yes

Reviewer #2: Yes

4. Is the manuscript presented in an intelligible fashion and written in standard English?

Reviewer #1: Yes

Reviewer #2: Yes

5. Review Comments to the Author

Reviewer #1: This manuscript presents an important dimension in the life of patients with chronic pain. Social support for the patients with chronic pain is part of the biopsychosocial model and addressing this topic provides the literature with more insights of this regard.

The manuscript is well-organised and well-written. From methodological perspective, this systematic review followed most of the conduction and reporting guidelines. However, there are few points to be considered:

1. the excluded full-text reports needs to be presented in a supplementary list. However, some modifications are needed as some reports should have been excluded in the title and abstract phase.

2. Assessing the quality of evidence using the GRADE is not very well matching this guideline in specific. We suggest either giving more details in the methods section about the tool and the table you created.

Your effort to produce this manuscript is highly appreciated

Reviewer #2: Review Report

Abstract:Lacks clarity and not compressive

Background:Lacks strength and focus as well as smooth transition

Methods: Incomplete and the dependent variable should be well operationalized

Results and discussion:Inadequately clear, brief and logical. The discussion lacks consistency throughout it's contents and proper explanation and references.

6. PLOS authors have the option to publish the peer review history of their article (what does this mean? ). If published, this will include your full peer review and any attached files.

**Do you want your identity to be public for this peer review?** For information about this choice, including consent withdrawal, please see our Privacy Policy .

Reviewer #1: No

Reviewer #2: No

- - - - -

---

## [Author Response · Author response to Decision Letter 1]

4 Dec 2024

Dear Editor,

Please find our response to the reviewers attached in the file inventory under the following name: "PLOS ONE - Rebuttal letter".

Kind regards,

Dr Rinaudo

---

## [Decision Letter · Decision Letter 1]

10 Jan 2025

PONE-D-24-21939R1Navigating the Biopsychosocial Landscape: A Systematic Review on the Association between Social Support and Chronic PainPLOS ONE

Dear Dr. Rinaudo,

Thank you for submitting your manuscript to PLOS ONE. After careful consideration, we feel that it has merit but does not fully meet PLOS ONE’s publication criteria as it currently stands. Therefore, we invite you to submit a revised version of the manuscript that addresses the points raised during the review process.

I was asked to handle this manuscript after an initial review, and I invited two new experts in the field to review the revised manuscript and the responses given to the original reviewers. Both reviewers suggest minor revisions, so please have a look at the changes suggested and respond to the new reviewers accordingly. Comments from PLOS Editorial Office: We note that one or more reviewers has recommended that you cite specific previously published works. As always, we recommend that you please review and evaluate the requested works to determine whether they are relevant and should be cited. It is not a requirement to cite these works. We appreciate your attention to this request.

We look forward to receiving your revised manuscript.

Kind regards,

Bianka Karshikoff, PhD

Academic Editor

PLOS ONE

Journal Requirements:

“This project has received funding from the European Union’s Horizon2020 research and innovation programme under grant agreement No 848068”

Please state what role the funders took in the study. If the funders had no role, please state:

If this statement is not correct you must amend it as needed. Please include this amended Role of Funder statement in your cover letter; we will change the online submission form on your behalf.

“We thank Marie Longton (UCLouvain) for providing help in assembling the research equation.

This project has received funding from the European Union’s Horizon2020 research and innovation program under grant agreement No 848068. This publication reflects only the authors' view and the European Commission is not responsible for any use that may be made of the information it contains.

C.M. Rinaudo and M. Van de Velde are joint first authors.”

“This project has received funding from the European Union’s Horizon2020 research and innovation programme under grant agreement No 848068”

5. Please include your tables as part of your main manuscript and remove the individual files. Please note that supplementary tables (should remain/ be uploaded) as separate "supporting information" files

Reviewers' comments:

Reviewer's Responses to Questions

**Comments to the Author**

1. If the authors have adequately addressed your comments raised in a previous round of review and you feel that this manuscript is now acceptable for publication, you may indicate that here to bypass the “Comments to the Author” section, enter your conflict of interest statement in the “Confidential to Editor” section, and submit your "Accept" recommendation.

Reviewer #3: All comments have been addressed

Reviewer #4: All comments have been addressed

2. Is the manuscript technically sound, and do the data support the conclusions?

Reviewer #3: Yes

Reviewer #4: Yes

3. Has the statistical analysis been performed appropriately and rigorously? 

Reviewer #3: Yes

Reviewer #4: Yes

4. Have the authors made all data underlying the findings in their manuscript fully available?

Reviewer #3: Yes

Reviewer #4: Yes

5. Is the manuscript presented in an intelligible fashion and written in standard English?

Reviewer #3: Yes

Reviewer #4: Yes

6. Review Comments to the Author

Reviewer #3: I support the publication of this revision, as the authors have performed a convincing study. I would like to see more discussions/future directions on how your findings could translate to social support based treatments/interventions. In my opinion, this is important for the clinical world.

Reviewer #4: Review

Navigating the Biopsychosocial Landscape: A Systematic Review on the Association between Social Support and Chronic Pain

ABSTRACT

Line 33-34: Indicate the Risk of bias tool utilised

TEXT

Line 40: Here you write “ most of the studies were cross-sectional” – in line 211 you write “The studies included in our review are exclusively observational”. Please change the sentences being consistent.

Line 48: “a survey showed that” … Consider indicating the argument of the survey to facilitate reading

Line 93: point at the end of the sentence.

Line 130-132: This part has to be moved to the discussion session

Line 133-136: The reference is old. Please specify whether something updated exists or whether this subdivision is still applicable today

Line 139: consider a recent SR (Bisconti M, Esposto M, Tamborrino A, Brindisino F, Giovannico G, Salvioli S. Is Social Support Associated With Clinical Outcomes in Adults With Nonspecific Chronic Low Back Pain? A Systematic Review. Clin J Pain. 2024 Oct 1;40(10):607-617. doi: 10.1097/AJP.0000000000001239. PMID: 39268726.)

Line 618: Consider to explain better in method how did you adapt the tool.

Very interesting. The discussion highlights the critical issues that research is currently facing regarding the issue of social aspects of pain.

Well-conducted methodological part, useful graphical representation of results.

7. PLOS authors have the option to publish the peer review history of their article (what does this mean? ). If published, this will include your full peer review and any attached files.

**Do you want your identity to be public for this peer review?** For information about this choice, including consent withdrawal, please see our Privacy Policy .

Reviewer #3: No

Reviewer #4: No

---

## [Author Response · Author response to Decision Letter 2]

16 Feb 2025

Dear Reviewers,

As requested, the manuscript has been reviewed accordingly to the Journal Requirements and your comments. The responses can be found in the following document "Response to Reviewers.docx". We thank you for you comments and positive review.

Kind regards,

Dr Rinaudo

---

## [Decision Letter · Decision Letter 2]

12 Mar 2025

Navigating the Biopsychosocial Landscape: A Systematic Review on the Association between Social Support and Chronic Pain

PONE-D-24-21939R2

Dear Dr. Rinaudo,

We’re pleased to inform you that your manuscript has been judged scientifically suitable for publication and will be formally accepted for publication once it meets all outstanding technical requirements.

Kind regards,

Bianka Karshikoff, PhD

Academic Editor

PLOS ONE

Additional Editor Comments (optional):

Reviewers' comments:

Reviewer's Responses to Questions

**Comments to the Author**

1. If the authors have adequately addressed your comments raised in a previous round of review and you feel that this manuscript is now acceptable for publication, you may indicate that here to bypass the “Comments to the Author” section, enter your conflict of interest statement in the “Confidential to Editor” section, and submit your "Accept" recommendation.

Reviewer #3: All comments have been addressed

Reviewer #4: All comments have been addressed

2. Is the manuscript technically sound, and do the data support the conclusions?

Reviewer #3: Yes

Reviewer #4: Yes

3. Has the statistical analysis been performed appropriately and rigorously? 

Reviewer #3: Yes

Reviewer #4: Yes

4. Have the authors made all data underlying the findings in their manuscript fully available?

Reviewer #3: Yes

Reviewer #4: Yes

5. Is the manuscript presented in an intelligible fashion and written in standard English?

Reviewer #3: Yes

Reviewer #4: Yes

6. Review Comments to the Author

Reviewer #3: The authors have addressed the concerns I raised in the last round. I have no more concerns and I want to congratulate the authors for such a massive review.

Reviewer #4: (No Response)

7. PLOS authors have the option to publish the peer review history of their article (what does this mean? ). If published, this will include your full peer review and any attached files.

**Do you want your identity to be public for this peer review?** For information about this choice, including consent withdrawal, please see our Privacy Policy .

Reviewer #3: **Yes: ** Xianwei Che

Reviewer #4: No

---

## [Editor Report · Acceptance letter]

PONE-D-24-21939R2

PLOS ONE

Dear Dr. Rinaudo,

I'm pleased to inform you that your manuscript has been deemed suitable for publication in PLOS ONE. Congratulations! Your manuscript is now being handed over to our production team.

Kind regards,

on behalf of

Dr. Bianka Karshikoff

Academic Editor

PLOS ONE